# Improving the Buffer Energy Absorption Characteristics of Movable Lander-Numerical and Experimental Studies

**DOI:** 10.3390/ma13153340

**Published:** 2020-07-27

**Authors:** Jinhua Zhou, Shan Jia, Jiacheng Qian, Meng Chen, Jinbao Chen

**Affiliations:** 1College of Astronautics, Nanjing University of Aeronautics and Astronautics, Nanjing 210016, China; jhzhou2018@163.com (J.Z.); nuaaqjc1994@nuaa.edu.cn (J.Q.); workmailcm@126.com (M.C.); chenjbao@nuaa.edu.cn (J.C.); 2In-orbit Control and Landing Laboratory, Nanjing University of Aeronautics and Astronautics, Nanjing 210016, China

**Keywords:** movable lander (ML), buffering performance, geometric parameters (GP), energy absorption (EA)

## Abstract

To improve the soft-landing crash performance of the movable lander (ML), this study presents an investigation of a newly designed gradual energy-absorbing structure subjected to impact loads using an ML for theoretical calculation and numerical simulations. In this work, we present a novel computational approach to optimizing the energy absorption (EA) of the ML. Our framework takes as inputting the geometrical parameter (GP) as well as EA. The finite element model of the HB1, HB2, and HB3 was established and effectively verified using numerical simulation and experimental data. The relationship between the GP of the buffer material and the EA was obtained through static experiment and impact experiment, and the cushioning performance of the lander was optimized according to the ML load mass, contact speed, and EA function. According to the optimization results, we chose an outer diameter of 240 mm, an inner diameter of 50 mm, heights of HB1 = 140 mm, HB2 = 110 mm, and HB3 = 225 as the collocation, and completed the numerical simulation of three different cases. By comparing the results of theoretical calculation and numerical simulation experiments, it can be found that the overload response rates of the main body in 4 type landing, 2-2 type landing, and 1-2-1 type landing are 4.72 G, 2.61 G, and 2.33 G, respectively. It also laid the foundation for the theoretical and methodological research of the ML and manned lander in the future.

## 1. Introduction

In the process of lunar exploration, only relying on the rover can no longer meet the requirements of lunar reconnaissance with heavy equipment in the future. Inspired by this, researchers have developed a movable lander (ML) [1], which can not only perform a soft landing on the lunar surface, but also complete walking. The process of creating new types of MLs, however, is notoriously challenging because their energy absorption (EA) capabilities is intimately related to their design. For this reason, creating new ML requires a great deal of experience, and is a largely manual and time-consuming task. This tedious and error-prone approach to creating the ML is unfortunately necessitated by the lack of formal models that can predict the complex interactions between the design of an ML and its ability to effectively serve its intended purpose.

Rather than relying on trial-and-error approaches, we seek to develop a computational model with the predictive power required to inform design decisions. To achieve this goal, we must establish a relationship between the aluminum honeycomb (AH) geometry parameters and EP of the ML. To this end, the effectiveness of the proposed method can be tested by EA characteristics under different landing environments [2,3], include 4 type landings on level ground, 2-2 type landing, and 1-2-1 type slope landings [4,5]. Generally, the cushioning performance of the lander aims to minimize the mass or volume of the cushioning material. The fast and elitist non-dominated sorting in genetic algorithms (NSGA-II) is used to figure out the geometric dimensions of different types of cushioning materials required, and the theoretical and simulation results are compared and verified [6,7]. In the past few decades, researchers have been working on developing mathematical models to complete the dynamics simulation of the lunar lander’s soft landing [8].

At present, many kinds of buffer structures have been widely used in industry and aerospace, such as airbag buffer [9], spring damping buffer [10], hydraulic buffer [11] and compressible material buffer [12]. During the study of the cushioning characteristics of the ML, both the airbag buffer and the spring damping buffer had been not conducive to the control and easy to cause the spacecraft to bounce and roll. The buffer fluid in the hydraulic buffer structure has been leaking in the past, which has limited its application for planetary exploration. Therefore, the compressible buffering method with simple structure, small mass, and not easy to bounce or roll is taken as the research object.

The lightweight and reliability of the lander has become the key technology to optimize the cushion performance. The AH structures are of demonstrative advantages of lightweight, excellent load bearing capacity and EA efficiency, which has been extensively used as energy absorbers in automotive, railway, naval, and aerospace industries. The experimental methods [13,14] analytical methods [15], and numerical methods [16,17] are generally used to test the mechanical properties of the honeycomb materials. Due to the good crushing effect of honeycomb on the outside, a large number of experimental studies have been conducted on its dynamic compressive strength, and it is found that the thickness [18], side length [19,20], and impact velocity [21] has the most obvious influence on the EA characteristics of the material.

The combination of experimental and Digital Image Correlation results worked excellently in the determination of global and local deformation mechanism of the honeycomb core (Khan [22]). A compression/shear coupling with transverse shear is introduced to predict the response of honeycomb sandwich structures to impact loads [23]. The dynamic impact crushing behavior of multi-layer honeycomb sandwich panels and the impact ensiled loading behavior of their material members is examined in this experimental investigation [24].

The main method of EA modeling of cushioning materials is the finite element method (FEM) and multi-body system dynamics method. Among them, the FEM can consider the nonlinear characteristics of geometric materials and contacts, so it can usually get more accurate results. One disadvantage of this method is that it is too time-consuming to analyze the multi-body dynamic model. Although the accuracy is not high, it can greatly improve the computational efficiency. In this paper, the physical parameters of the buffer material were obtained by using the finite element method. The multi-body system dynamics method was selected for the study of the soft-landing characteristics of the whole machine, and was compared with the theoretical calculation results [25].

Since the 1960s, research on the buffer characteristics of the lunar lander during landing has been widely carried out. A comparative study of the stability of the three-legged lander and the four-legged lander by Lavender, found that the diameter envelope of the three-legged lander was slightly larger than that of the four-legged lander, but the three-legged lander had a smaller overall weight [26]. Robert and Warner propose a method to solve the elastic rebound problem of the lander by improving the buffer materials [27]. The theoretical model and results of a parametric study had been given in terms of ground slope (5°, 7°, and 15°) [28]. Blanchard carried out impact simulation test on the full-size lunar lander model under lunar gravity with the simulation analysis method, and obtained good test results [29]. Hilderman et al. have verified the cushioning reliability of landing by studying the EA characteristics of the landing buffer under various harsh landing conditions [30]. William introduced the development of lunar module landing gear subsystem in the Apollo 11 lunar landing mission in detail, and verified the lander test analysis method under different working conditions [31].

Lunar landers around the world need a lot of experiments before they can be launched. There are too many unknown difficulties in the real prototype experiment due to its shortcomings such as long manufacturing cycle, difficult data acquisition and large assembly error. The numerical simulation technology is widely used in the aerospace field, due to its brand-new research and development mode, which has the characteristics of low research and development cost, short research and development cycle and high product quality. Due to the soft-landing characteristics and overload rate of the ML, its safety is greatly affected. At present, there is only research on the soft-landing characteristics of the traditional lunar lander, and there is no research data on the buffering characteristics of the ML.

Additionally, the analysis of landing overload performance has also attracted great attention from the literature. Jones and Hinchey [32] and Kushida [33] have studied the impact of landing environment on airframe overload response, providing guidance for the study of buffering characteristics. In order to study the impact of different influencing factors on the overload response of the Lunar lander, Alderson and Wells [34] created mathematical models and computer programs. In reference [35], a 1/6-scale dynamic model of a lunar module type of spacecraft has been tested to determine the overturning stability boundaries for landings on nonlevel surfaces.

The reliability of a soft landing is a very important technical index in the process of lander development, for example, the design process of the Apollo lander [36]. Research objectives of lander mainly focus on the EA characteristics and conceptual design of soft landing [37]. In addition, methods to study the EA characteristics of the lander can be generally divided into three categories [38], namely, theoretical analysis method [39], simulation analysis method [40], and experimental method. In particular, simulation method is commonly used at present, but it needs to find optimal buffer material collocation through repeated iteration and simulation for many times. Therefore, in order to obtain the best geometric parameters (GP) of buffer materials, it is necessary to combine mathematical calculation method to optimize the combination and collocation mode of buffer materials and quickly find the optimization method of the EA on the basis of theoretical research.

It can be seen from previous research that EA of AH and other buffer materials had always depended on their GP. In order to estimate the EA of the AH, the load of the lander, the lateral velocity, horizontal velocity, and landing environment should be considered. On this basis, the influence of cushion material on the overload coefficient of the body under the impact load of the ML is studied. In this paper, a more effective method using theoretical derivation, simulation experiment, and NSGA-II optimization method is proposed.

This article will provide the theoretical basis and technical support for the design of the ML’s buffer structure based on the theoretical analysis and numerical simulation, and by optimizing the design of the EA characteristics of the aluminum honeycomb. Numerical simulation and virtual prototype technology have been used to design a reliable EA buffer structure with lower cost in a short development cycle. This research result will play a certain role in promoting the development of cushion technology for ML.

## 2. Introduction to the ML

This paper designs a novel movable lunar landing mechanism. The mechanism consists of four identical landing legs evenly mounted around the body of the ML. The landing leg is composed of a primary strut, two secondary struts, a swinging beam, a main leg, and a foot pad. The primary strut and secondary strut adopt the same structure including outer tube, inner tubes, driving mechanism and buffer materials, as shown in Figure 1.

The structure of the primary strut and secondary strut is composed of inter tube, outer tube, aluminum honeycomb-I, aluminum honeycomb-II, driving mechanism, ball screw, screw nut, and push-pull locking mechanism. The inter tube of the primary strut is connected with the rotary joint, and the inter tube of the secondary strut is connected with the spherical joint, which is also the only difference between the primary strut and the secondary strut.

The primary strut and the swinging beam are connected by a rotating joint. The primary strut and the swinging beam are connected to the body through the rotary joint. The joints of the main leg connecting the swinging beam and the foot pad are a universal joint and a spherical joint, respectively. The two secondary struts and the main leg are connected by the ball joint. The main body and the auxiliary pillar are connected by a universal joint. The primary struts are crushed by the internal AH to buffer the vertical impact of the main body.

The secondary strut needs to withstand the vertical and lateral swing restraint forces exerted by the main leg, and the restraint force is achieved by the compression EA of the AH material inside the secondary strut.

Working Principle of the ML:

In the initial state, the buffer mechanism and the drive mechanism is independent of each other, the inter tube is locked with the push-pull locking mechanism, and the push-pull locking mechanism is installed in the middle of the aluminum honeycomb-I and aluminum honeycomb-II. At the same time, a screw nut is located on the left side of the ball screw, as shown in Figure 1. No connection has been established between the lead screw nut and the inner tube. During the soft landing on the lunar surface, due to the impact of different landing speeds and irregular lunar surface environment, the stress direction of the four landing legs and landing inclination Angle of the ML has many uncertainties. After landing, the main leg and swinging beam transmit the contact force between the foot pad and the lunar surface to the primary struts and secondary struts through the inter tube. Because the direction of the force is different, each primary strut and secondary strut are subjected to tension or pressure, and the aluminum honeycomb-I or aluminum honeycomb-II is compressed by the push-pull locking mechanism to achieve cushioning.

After the soft-landing mission is completed, the ML needs to complete attitude adjustment or walking preparation work. At this time, the driving mechanism starts to move and drives the ball screw to rotate. Because the circumferential freedom of screw nut is limited, the screw nut can only move horizontally. When the screw nut moves to the locking position of the inter tube, it is locked together with the inter tube, and the lock between the inter tube and the push-pull locking mechanism is released. The screw nut drives the inner tube of the primary struts and secondary struts to drive the swing beam and the main leg to swing, and finally achieves movability.

## 3. Algorithm

TheNSGA-II is an advanced multi-objective optimization method based on evolutionary algorithms. This algorithm not only reduces the complexity of non-inferior sorting genetic algorithm, but also has the advantages of faster running speed and better convergence of the solution set. NSGA-II can find the Pareto optimal solution set that makes each objective function as large as possible (or as small as possible), and thus provides an effective tool for balancing between each objective function.

When optimizing the EA of the ML, it is described by the GP and EA parameters of the material. Our goal is to study an optimization design method, based on the relationship between the EA and the GP of the material satisfying the constraint equation, and link the physical properties of the material with its EA. In some cases, we can calculate how to change the subset of GP of the buffer material in response to changes in its EA parameters while maintaining constraints. The subset of the GP of the buffer material can also be changed according to the buffer EA requirements, as shown in Figure 2.

In Figure 2, we optimized the EA parameters and GP (such as material length, cross-sectional area, etc.) of the buffer material according to the requirements of the ML’s buffer EA. Firstly, we will get the EA parameters and GP requirement of the buffer material through theoretical calculation and simulation. If the material parameters meet the material’s mass constraint, the GP constraint and EA constraint. Next, we will take the minimum total mass of the material, the VEA and MEA as the objective function. Finally, the NSGA-II is used to obtain optimization results that meet the design requirements. This optimization method provides a new research direction for the research of the cushioning EA of the ML: the user can calculate the optimal combination of the physical parameters of the buffering materials and their EA changes within the constraints.

In this paper, the optimization method of combination mode of energy absorbing materials needs to be completed by theoretical derivation, simulation experiment and the NSGA-II [41,42,43]. Moreover, the NSGA-IIcan obtain the non-inferior solution set through Pareto superiority method. All Pareto sets are feasible optimization schemes. The multi-objective optimization procedure is the optimization of a solution set. Even if the individual difference between generations is less than the convergence threshold, the optimal individuals that selected from the convergent groups of different generations according to the same requirement are still different. In order to reduce this difference, i.e., to improve the optimization accuracy, a large number of iterative calculations are often required. Generally, the expression of multi-objective problem of minimization is as follows:(1)$$Min:f\left(x\right)=\left\{f_{1}\left(x\right),f_{2}\left(x\right)\ldots f_{m}\left(x\right)\right\}\;Subject\;to:e\left(x\right)=\left(e_{1}\left(x\right),e_{2}\left(x\right)\ldots e_{m}\left(x\right)\right)\;Where:\;x=\left(f_{1},f_{2},\ldots f_{n}\right)\in X\;y=\left(y_{1},y_{2},\ldots y_{n}\right)\in Y$$
where f(x) denotes the target vector of all design objectives; *m* denotes the number of target response functions;$\;f_{i}\left(x\right)$ denotes the ith design goal; $e_{i}\left(x\right)$ denotes the ith constraint function; X denotes the domain of variables; and *n* denotes variation coefficient

In the optimization of buffering EA parameters of the ML. The mathematical definition of Pareto law is: for any x∈X, there is no $x^{\prime }\in X$. Such that $f\left(x\right)=\left\{f_{1}\left(x\right),f_{2}\left(x\right)\ldots f_{m}\left(x\right)\right\}$ is better than $f\left(x\right)=\left\{f_{1}\left(x\right),f_{2}\left(x\right)\ldots f_{m}\left(x\right)\right\}$ and x is the Pareto optimal solution on  X. Where $X_{1}$ is better than $X_{2}$. The NSGA-II is used to solve the Pareto optimal solution set in order to obtain the final solution of the buffer material’s GP.

## 4. Materials and Methods

### 4.1. Quasi-Static Model of the Material is Established

#### 4.1.1. Stress Model Hypothesis

In the design of the buffer device, due to the influence of the material EA characteristics and the direction of the load, we aim at the effective mass and the maximum EA in the volume.

Based on the buffer materials of Chang 'e-3 and Chang 'e-4, the hexagon honeycomb structure and Y-cell in the honeycomb structure was selected for the study, as shown in Figure 3. The simplified hexagonal metal honeycomb structure is represented by two Y-cell with a thickness of t and a length of l/2 and a Y-cell with a thickness of  2t  and a length of  w/2. As shown in Figure 3a. The angle between the two adjacent sides of the hexagonal metal honeycomb cell is α, and the simplified Y- cell cross-section is shown in Figure 3b.

To determine the static pressure characteristics of the honeycomb structure, we make the following assumptions: (1) the honeycomb matrix material has good plasticity, which is regarded as an ideal elastoplastic material and (2) the effect of bonding on mechanical properties of honeycomb structures is not considered.

In order to obtain the compression performance of the three honeycomb materials under out-of-plane static compression or impact load, based on the symmetric structure of the honeycomb structures and law of energy conservation. Based on this, average compressive stress constitutive equation under out-of-plane compression is deduced with Mises yield criterion and Tresca yield criterion. Considering the effect of strain rate on its mechanical characteristics based on the Cowper-Symonds model, dynamic average compressive stress calculating equation is deduced. In order to reduce the influence of the honeycomb structure of the same specification due to different external dimensions on the average force, in this paper, the average stress of the honeycomb structure is used instead of the average load. Using the method in [44,45], calculated the percentage of Y-cell in Figure 3b. The area of Y-Cell can be obtained by Equation (2).
(2)S=lcosa(w+lsina)

#### 4.1.2. Quasi Static Compression Experiment

Three kinds of buffer materials were selected and installed in the main and auxiliary pillars in the matching way as shown in Figure 2. Experiment parameters of buffer materials were shown in Table 1.

In order to obtain more accurate compression rate parameters of three aluminium honeycomb materials (HB1, HB2, and HB3) under static pressure. According to the geometric parameters of HB1, HB2, and HB3 in Table 1, their shell model was established by using commercial software Workbench. The material density was set to 2730, 2730, and 2680 kg/m^3^, respectively. The tensile strength was set to 150, 150, and 200 MPa, respectively. The yield strength was set to 115, 115, and 65 MPa, respectively. In order to capture the mechanical properties of the honeycomb under large plastic deformation, the type of elements for the three honeycomb materials is modelled with tetrahedral mesh elements with 4 nodes. The element size is set to 0.02 mm. The number of finite elements was 1,519,496, 2,269,506 and 2,198,152, respectively. The reference point method is used to determine the boundary conditions of honeycomb materials. In order to avoid the honeycomb structure buckling during the compression process. A fixed constraint was applied to the bottom of the element model and a constant pressure of 50 KN was applied to the upper surface, and the simulation time is set to 1500 s. Simulation results of static pressure are shown in Figure 4 and Table 1.

In order to obtain the accurate compression rate of the three materials in Table 1, data support is provided for the design of the GP of the buffer material. First, the three AH materials were subjected to a static pressure simulation experiment. The experiment results are shown in Figure 4 and Table 2. Then, the axial quasi-static compression experiment is carried out using the three materials of the universal MTS testing machine as shown in Figure 5 and Table 2, the testing machine adopts displacement driving test, the maximal force of the testing machine is 50 KN and the speed is 5 mm/min. The static pressure experiment results are shown in Figure 6. During the experiment, all the experiment pieces are concentrated between the loading platform and the stationary platform of the experimenting machine. The data collection system of the MTS machine is used to automatically record the crushing load and crushing displacement. At the same time, the crushing process is recorded with a camera every 30 s.

Under the condition of static pressure load, simulation experiment and static pressure test were carried out on the buffer material, and the load-displacement curves of HB1, HB2, and HB3 were obtained, as shown in Figure 7a–c. It can be observed that simulation results agree with the experimental results well. The difference between the experimental and numerical curves may be caused by the rupture of the material in the experimental test while the numerical models did not take into account the failure of material. Through experiments we have obtained the average load of HB1, HB2, and HB3 are 6.25, 8.27, and 9.55 KN, respectively.

### 4.2. EA Characteristics

The landing process of the ML needs to bear a large impact load, and its mass and volume need to be strictly controlled. Evaluate the mass and volume of the buffer device and define the maximum mass specific energy absorption (MSEA) and volume specific energy absorption (VSEA) of the material:(3)$$\{\begin{array}{c} SEA_{m}\frac{W_{total}}{m}=\frac{\int_{0}^{\delta }F\left(\delta \right)d\delta }{m}\\ SEA_{v}\frac{W_{total}}{V}=\frac{\int_{0}^{\delta }F\left(\delta \right)d\delta }{V} \end{array}$$
where, $W_{total}$ denotes the total EA by the material; m denotes the mass of buffer material; δ denotes the compression deformation; and F(δ) denotes the load that the material bears when it is deformed to δ; V denotes the Buffer volume.

The VSEA of the hexagonal honeycomb structure was analyzed. The Y-cells of the honeycomb structure were extracted for study. The VSEA of the hexagonal honeycomb structure was calculated.

In Reference [46], the theoretical formula of dynamic mean stress and ultimate strain of hexagonal honeycomb structures under impact load based on Mises strength criterion is derived. As shown in Equations (4) and (5):(4)$$\sigma_{m}^{d}=\frac{\frac{\sqrt{3}}{3}\pi \sigma_{0}^{\;}t^{2}\left(l+2w\right)}{k\sqrt{\frac{\sqrt{3}}{6}\pi t(l+2w})(lcosa\left(w+lsina\right)}[1+{\left(\frac{v_{0}}{8k_{1}D\sqrt{\frac{\sqrt{3}}{6}\pi t(l+2w})}\right)}^{\frac{1}{P}}$$
(5)$$\varepsilon_{D}=1-k_{1}\frac{t}{\sqrt{\frac{\sqrt{3}}{6}\pi t(l+2w})}$$
where, $\sigma_{0}^{\;}$ denotes the initial yield strength of the honeycomb matrix material; k denotes the compression ratio, that is the effective compression length coefficient, as shown in Table 2. $\alpha =30^{{}^{\circ}}$, l=w, D denotes the Strain rate sensitivity coefficient of material; $v_{0}$ denotes the initial velocity of impact; P denotes the strain rate sensitivity coefficient of material; and $k_{1}$ denotes the height compensation factor. Based on the difference between the effective compression height and the theoretical compression height is calculated based on simulation and experiment, choose between 1.4~1.6, we choose $k_{1}=1.6$ [47].

The maximum effective EA of the Y-cell was:(6)$$W_{Y}=10^{-3}\sigma_{m}^{d}S\varepsilon_{D}L$$
where, L denotes the height of the Y-cell.

Substituting Equations (2), (4), and (5) into Equation (6), the maximum effective EA of Y-cell can be obtained as follows:(7)$$W_{Y}=\frac{20^{-3}\sigma_{0}t}{k}\left[1+{\left(\frac{v_{0}}{8k_{1}D\sqrt{\frac{\sqrt{3}}{6}\pi t\left(l+2w\right)}}\right)}^{\frac{1}{p}}\right]\left(\sqrt{\frac{\sqrt{3}}{6}\pi t\left(l+2w\right)}-k_{1}t\right)L$$

The mass and volume of Y-cell are shown in Equation (8):(8)$$\{\begin{array}{c} m=10^{-9}\rho_{h}V=10^{-9}\left(wt+lt\right)\rho_{b}L\\ V=10^{-9}SL=10^{-9}lcosa\left(w+lsina\right)L \end{array}$$
where, $\rho_{b}$ denotes the base material density, corresponds to the actual material density of the aluminum honeycomb material, $\rho_{h}$ denotes the material structure density, the density of solid material of the same mass, and volume as aluminum honeycomb material.  S  denotes cross section area of Y-cell.

Combined Equations (5), (7), and (8). The theoretical calculation models of MSEA and VSEA are obtained:(9)$$\{\begin{array}{c} SEA_{m}=\frac{W_{total}}{m}=\frac{20^{6}\sigma_{0}t}{k}\left[1+{\left(\frac{v_{0}}{8k_{1}D\sqrt{\frac{\sqrt{3}}{6}\pi tl}}\right)}^{\frac{1}{p}}\right]\left(\frac{\sqrt{\frac{\sqrt{3}}{2}\left(\pi tl\right)}-k_{1}t}{tl\rho_{b}}\right)\;\\ SEA_{v}=\frac{W_{total}}{V}=\frac{20^{6}\sigma_{0}t[\sqrt{\frac{\sqrt{3}}{6}\pi t\left(l+2w\right)}-k_{1}t}{k\left[lcosa\left(w+lsina\right)\right]}\left[1+{\left(\frac{v_{0}}{8k_{1}D\sqrt{\frac{\sqrt{3}}{6}\pi t\left(l+2w\right)}}\right)}^{\frac{1}{p}}\right] \end{array}$$

### 4.3. Dynamic Impact Simulation

Because the precision transmission mechanism is installed inside the buffer material, its structure is designed as a ring structure. We combine the material parameters in Table 1 to conduct simulation experiments in the collocation mode shown in Figure 8.

In order to reduce the overload response of the buffer, the impact force of each landing leg of the ML under different landing conditions is different. In order to ensure the reliability of the cushioning material, we need to consider the landing buffer under different conditions, and select the following Table 3. The experiment collocation method completes the simulation experiment and impact experiment.

In Table 3, considering the different landing environment of the ML, the impact force of each landing leg is greatly different. When four legs land simultaneously, the maximum buffer EA of a single leg can be equivalent to the impact energy with a load mass of 300 kg. During the 2-2 type landing, the maximum buffer EA of a single leg can be equivalent to the impact energy with a load mass of 500 kg. During 1-2-1 landing, the maximum buffer EA of a single leg can be equivalent to the impact EA when the load mass is 700 kg. Therefore, we chose 300, 500, and 700 kg as the load masses.

In the simulation experiment, different load masses are used to impact different lengths of material combinations, as shown in Figure 9. The combined parameters of the materials are shown in Table 3.

In this paper, ANSYS software is used as the pre-processing software for finite element analysis, and LS-DYNA software solves the impact of AH materials. In order to improve the computing speed, the grid cell size of HB1, HB2, and HB3 were set to 0.5 mm. Honeycomb of Nonlinear-Foam Material Models is used as a material model in HB1, HB2, and HB3. Material parameters are shown in Table 1. Density: set to 30, 36, and 50 kg/m^3^, respectively. Initial yield strength: set to 130, 135, and 135 MPa, respectively. Elasticity modulus: set to 40.39, 41.02, and 41.73 GPa, respectively. Poisson’s ratio: Set both to 0.33 Viscous damping coefficient set both to 0.2. The type of elements was selected for hexahedral mesh-Hex/Wedge. As shown in Figure 9. The buffer material model was placed between the rigid mass block and the fixed surface, and the bottom HB3 material was fixed. The reference point method [48] is adopted to determine the boundary conditions and initial velocity of the impact mass block. All degrees of freedom are limited except for translation along the direction of collision. When three material models in series are impacted by a mass block, in order to prevent contact between each surface of the model from penetrating, the contact type was selected to be single-side automatic contact. In order to ensure that the simulation results of metal honeycomb impact are close to the real results, the dynamic friction coefficient between the mass block and the aluminum honeycomb material is set to 0.2.

The ratio of the lander’s opposite acceleration to the earth’s gravitational acceleration (G) is called the overload rate. $G=9.8m/s^{2}$. During the soft landing of lunar lander, the maximum peak acceleration has a great influence on the safety of precision instruments and astronauts. In the aerospace field, the overload rate of the lander has been used as a key index to evaluate its safety performance.

It can be seen from Figure 10, when the load mass is 300 kg, the overload rate is 5.2 G. With the increase of the load mass to 500 and 700 kg, the total length of the buffer material is 450 and 600 mm, respectively. The overload rate was reduced to 3.2 and 3 G.

In Figure 11, when the load mass is 300 kg, the three materials are completely compressed. With the increase in the density of HB1, HB2, and HB3, the EA of HB1, HB2, and HB3 increase continuously. When the load mass is 500 kg, the total length of the three buffer materials increased in equal proportion, HB1 and HB2 are all compressed. The HB3 material was not completely compressed, resulting in the energy absorbed by HB2 and HB3 being similar. When the load mass was 700 kg, HB1 and HB2 were all compressed. HB3 absorbed less energy than HB2 because of the reduced ratio of length to which HB3 was compressed.

It can be seen from Figure 12, the EA conditions and matching methods of the three materials meet the requirements of the experimental design. After the HB1 buffer EA is completed, HB2 starts to absorb energy, and finally the HB3 material completes the buffer. Meet the design requirements.

In order to verify the feasibility of the proposed material combination, an experimental verification was carried out on a 300 kg load and 300 mm buffer material. We selected the CL-100 impact experiment bench to complete the impact experiment. It can be seen from Figure 13 that the material did not collapse locally after the impact. In Figure 14, the maximum overload rate of the simulation experiment is 5.3 G, and the maximum overload of the impact experiment is 6.4 G. The overload rate of the simulation is lower than that of the actual impact experiment, but it has the same trend. The matching method of the buffer experiment meets the requirements of buffer EA. We calculated the theoretical and simulated values of the MSEA and VSEA of the buffer material, as shown in Table 4.

In Figure 14, we can see that the variation trend of the numerical simulation results and the experimental results of the overload rate curve is similar. In the actual experiment, due to the friction between the load mass and the guide rail, the vibration between the test bed and the ground, and the sensitivity error of the sensor, there is a fluctuation error between the overload rate and the numerical simulation results. However, the experimental results meet the requirements of the impact test.

## 5. EA Optimization

When optimizing the EA characteristics of the aluminum honeycomb structure of the ML, we take the maximum VSEA and maximum MSEA of the three materials as optimization objectives, the mass and volume of aluminum honeycomb materials as design variables. The optimization goal of this paper is to reduce the overall mass and volume of the buffer material as much as possible. In order to further improve the EA characteristics of the buffer material. As the outer and inner diameters of the buffer materials are restricted by the structural parameters of the landing leg and the internal drive system, the three target functions of maximum VSEA $f_{1}\left(x\right)$, maximum MSEA $f_{2}\left(x\right)$, and maximum mass $f_{2}\left(x\right)$ of the three materials need to be evaluated.

The NSGA-II is an advanced multi-objective optimization method based on EA. This algorithm not only reduces the complexity of non-inferior sorting genetic algorithm, but also has the advantages of faster running speed and better convergence of solution set. The NSGA-II can obtain the optimal frontier of multi-objective optimization model efficiently by the improved fast non-dominated ranking and elite selection strategy without any parameter adjustment.

According to the model in Section 3, the search of volume, mass and energy optimal involved multiple objectives and constraints which exist a complex coupled relation and a serious non-linearity. So, the NSGA-II will be used to search the optimal solution. However, the multi-objective optimization is an optimization procedure of solution set, and the solution space is considerably large. As a result, the individuals of each generation are hard to be exactly the same, and the algorithm is difficult to achieve the absolutely convergence. In general, the inter-generational differences of individuals are used to indicate the convergence of the algorithm, and the choice of the optimal solution is by selecting the individual from the obtained optimal frontier solution set according to the requirement. Therefore, lots of iterative calculations are often required to improve the optimization accuracy. In addition, even if the individual difference between generations is less than the convergence threshold, there are still differences between the optimal individuals which selected from the convergent groups of different generations according to the same requirement. Therefore, it will easily lead to poor optimization accuracy or lots of redundant iterative operations when determining the convergence of the algorithm only based on the convergence threshold. In order to improve these disadvantages, the expected solution preserving strategy is introduced to the NSGA-II algorithm. That is, according to the expected optimization requirement, reserving the relative optimal solution in each generation (the expected solution) to the next generation and taking the stability of the expected solution as the index of convergence.

In the cushioning process, the total EA of the cushioning material is an important indicator to measure the EA of the material, the energy absorbed per unit mass (volume) is an important indicator of lightweight design and miniaturized design of buffer materials. Therefore, the objective function as shown in Equation (10) is created in order to obtain efficient and stable applications.
(10)$$\{\begin{array}{c} max\;f_{1}\left(x\right)=\overset{m}{\underset{i=1}{\sum }}SEA_{vi}v_{i}{}_{\;}\\ \begin{array}{c} max\;f_{2}\left(x\right)=\overset{m}{\underset{i=1}{\sum }}SEA_{mi}v_{i}\\ min\;f_{3}\left(x\right)=\overset{m}{\underset{i=1}{\sum }}\rho_{vi}v_{i} \end{array} \end{array}i\in 1,2,\ldots ,m$$
(11)$$s.t.\{\begin{array}{c} v_{min}\begin{array}{c} \leq v_{1}+v_{2}+v_{3} \end{array}\\ \begin{array}{c} v_{1}+v_{2}+v_{3}\leq v_{max}\\ v_{1},v_{2},v_{3}\geq 0\\ m_{min}\begin{array}{c} \leq m_{1}+m_{2}+m_{3} \end{array}\\ m_{1}+m_{2}+m_{3}\leq m_{max}\\ m_{1},m_{2},m\geq 0 \end{array} \end{array}$$

According to Equation (10), the sub-objective $f_{3}\left(x\right)$ is inversely proportional to $f_{1}\left(x\right)$ and $f_{2}\left(x\right)$. The $f_{3}\left(x\right)$ directly reflects the total mass of the buffer material.

Setting the population size ($N_{p}$) to 200, the maximum number of generations to 300, the crossover probability to 0.8 and the mutation probability to 0.1. The trend of the standard deviation for every 30 generations that calculated by the average values of the objective function which corresponding to the individuals in each generation is shown in Figure 15.

## 6. Numerical Modeling and Validation

### 6.1. Parameter Setting

The GP of the AH is used as design variables. The transmission mechanism and structural strength inside the strut need to be considered. The inner diameter of the AH is restricted by the precision screw (design diameter 50 mm), and the outer diameter design interval of the material is 60 to 280 mm and since the cross-sectional areas of the three buffer materials are the same, we will take the maximum total EA, the minimum volume and the minimum mass as the main optimization goals.

Combining with Equation (10), we take the design of the single-leg primary strut to absorb 7 kJ as the maximum energy constraint. We obtained the parameters of the three AH materials after rounding, the outer diameter and inner diameter were 240 and 50 mm, respectively. The theoretically calculated compression buffer lengths of the three materials are HB1 = 118 mm, HB2 = 94 mm, and HB3 = 184 mm, respectively. According to the different simulation compression rates of the three materials in Table 2, we calculated the lengths of the three materials as follows: HB1 = 140 mm, HB2 = 110 mm, and HB3 = 225 mm. Based on this combination of parameters, the ML Numerical simulation experiments of buffer characteristics. The simulation parameters of the ML mechanism and the material property parameters of the landing environment are shown in Table 5 and Table 6.

### 6.2. Constraints and Initial Condition Settings

When using multi-body system dynamics modeling, constraints and initial condition settings are one of the key steps in the landing simulation process. It is not only necessary to add friction constraints to each rotation and movement joint. In order to simplify the model, we also ignore the detailed model of each leg and use a simplified multi-body system dynamics method for simulation.

We compared the simulation results with the theoretical calculation and defined the interaction between the inner tube and the equivalent buffer, the joint angle of the motion joint, foot pad, and the landing surface. The friction characteristics of all contact points are defined as Coulomb friction. In the commercial software ADAMS, the method of setting friction parameters between two moving parts only needs to create friction in the connectors, without enforce a specific value of friction on a material surface. The friction parameter is set as shown in Table 7. The landing conditions are shown in Figure 16, and the landing condition parameters are set as shown in Table 8.

### 6.3. Simulation Results and Validation

The ML was numerically simulated according to three different working conditions in Table 8, and the buffering performance and EA results of case 1, case 2, and case 3 is analyzed respectively.

In case 1, the overload rate curve of the ML body is shown in Figure 17. When the four legs land at the same time, the buffer time is between 0.25 and 0.9 s. At this time, the maximum overload rate of the ML is 4.72 G. It meets the design requirements for overload rate of buffer landing. In order to verify the reliability of the designed structure, the theoretical value of the EA of the primary strut and secondary strut is compared with the simulation. As shown in Figure 18, after comparison, it is found that the theoretical calculation of the four primary struts. The EA is kept at about 4000 J, and the theoretical EA of the eight secondary struts is about 600 J. Energy loss is due to mechanical joint friction. There are some errors between theoretical EA and numerical simulation EA.

Case 2 is a 2-2 type landing, Leg1 and Leg2 first landed at the same time, then Leg3 and Leg4 landed at the same time. As can be seen from Figure 19, the main body’s buffer time is between 0.25–1.65 s, and the overload rate appears two similar peaks are 2.61 and 2.43 G, respectively. The overload factor meets the design requirements. At the same time, in order to verify whether the buffer structure is reliable, the theoretical and simulated EA of the AH material inside the primary strut and secondary struts are compared and analyzed. In Figure 20, we found that the theoretical EA of the primary struts of Leg1 and Leg2 is 4500 J, the theoretical EA of Leg3 and Leg4 primary struts is 2800 J, and the theoretical EA of secondary struts 1 and secondary struts 2 of Leg1 and Leg2 is 630 J. The theoretical EA of the secondary struts 1 and 2 of Leg3 and Leg4 is 500 J.

Case 3 is a 1-2-1 type landing, first, Leg1 landing, second, Leg2 and Leg4 landing at the same time, and finally, Leg3 landing, from Figure 21 we can see that the buffer time is actually between 0.22–1.1 s. The overload rate has three peaks, which are 1.31, 2.33, and 1.84 G, respectively. The maximum overload rate occurs when Leg2 and Leg4 land, because the two legs landed at the same time and the body’s center of mass was subjected to a large counterforce. Comparing Figure 22, we can see that the theoretical EA of Leg1 primary strut is 6220 J, the theoretical EA of Leg2 and Leg4 primary strut is 3800 J, and the EA of Leg3 primary strut is 2500 J. Due to the deflection of the ML, all the main legs have different side swing. The theoretical EA of the secondary struts 1 and 2 of Leg1 is 700 J, the theoretical EA of the secondary struts 1 and secondary struts 2 of Leg2 and Leg4 is 500–600 J, and theoretical EA of the secondary struts 1 and secondary struts 2 of Leg3 is 300 J.

## 7. Conclusions

In the design of soft landing and the prediction of soft-landing performance, impact dynamics analysis during landing is one of the most important tasks. When ML lands, the energy absorption characteristics of the cushioning material have many uncertainties. This paper uses theoretical calculation, algorithm optimization, and computer simulation technology to study the EA characteristics of the buffer. In this research, a new-type composite energy absorbing structure was designed by combining the characteristics of AH structures. The structure designed was applied to the buffer for the ML, in order to realize the design of EA functions in the under-frame structure of an ML structure. The following conclusions may be drawn as a result of this research:(1)The axial crushing characteristics of three different AH materials—HB1, HB2, and HB3—under static load and dynamic load were studied. Using the NSGA-II to solve a variety of different GP and EA parameter combination. We get the combination of maximum EA and minimum total mass parameters of the buffer material. An equivalent solid model of the honeycomb structure was established to effectively bring about the numerical simulation of the crushing of the honeycomb structure. The comparison suggested that the numerical simulation results were basically consistent with compression test data. This outcome revealed that the numerical model of the AH structure was effectively verified by test evidence.(2)The EA structures using three types of honeycomb structures (HB1, HB2, and HB3) were assessed by numerical simulation of their crushing behavior. It can be seen from dynamic simulation results that the EA process is divided into three stages, which corresponded to the three-level structure of the EA structure combination. In the whole crushing process, the whole structure is deformed step by step according to the process of energy consumption design. The relationship between the geometric structure of the cushioning material and the EA parameters is obtained through simulated EA experiments and dynamic impact experiments.(3)The dynamic simulation model of the multi-body system of the ML was established and verified by simulation experiments. Three different landing conditions are used to verify whether the combination of cushioning materials selected by the ML is suitable for the worst landing conditions. The greater the combined length of the honeycomb, the greater the contribution of the honeycomb to the entire EA structure. Taking the EA of three different landing cases as examples, the maximum overload rate obtained under simulation under different working conditions is: 4.72, 2.61 and 2.33 G. This model allows us to easily evaluate the landing performance of the ML, and can save a lot of difficult and expensive landing experiments.

## Figures and Tables

**Figure 1 materials-13-03340-f001:**
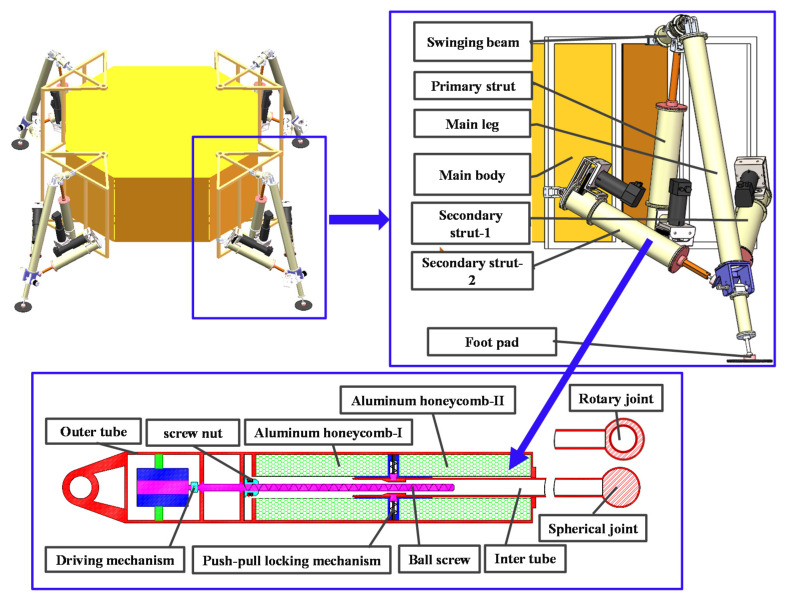
Geometrical description of the movable lander (ML).

**Figure 2 materials-13-03340-f002:**
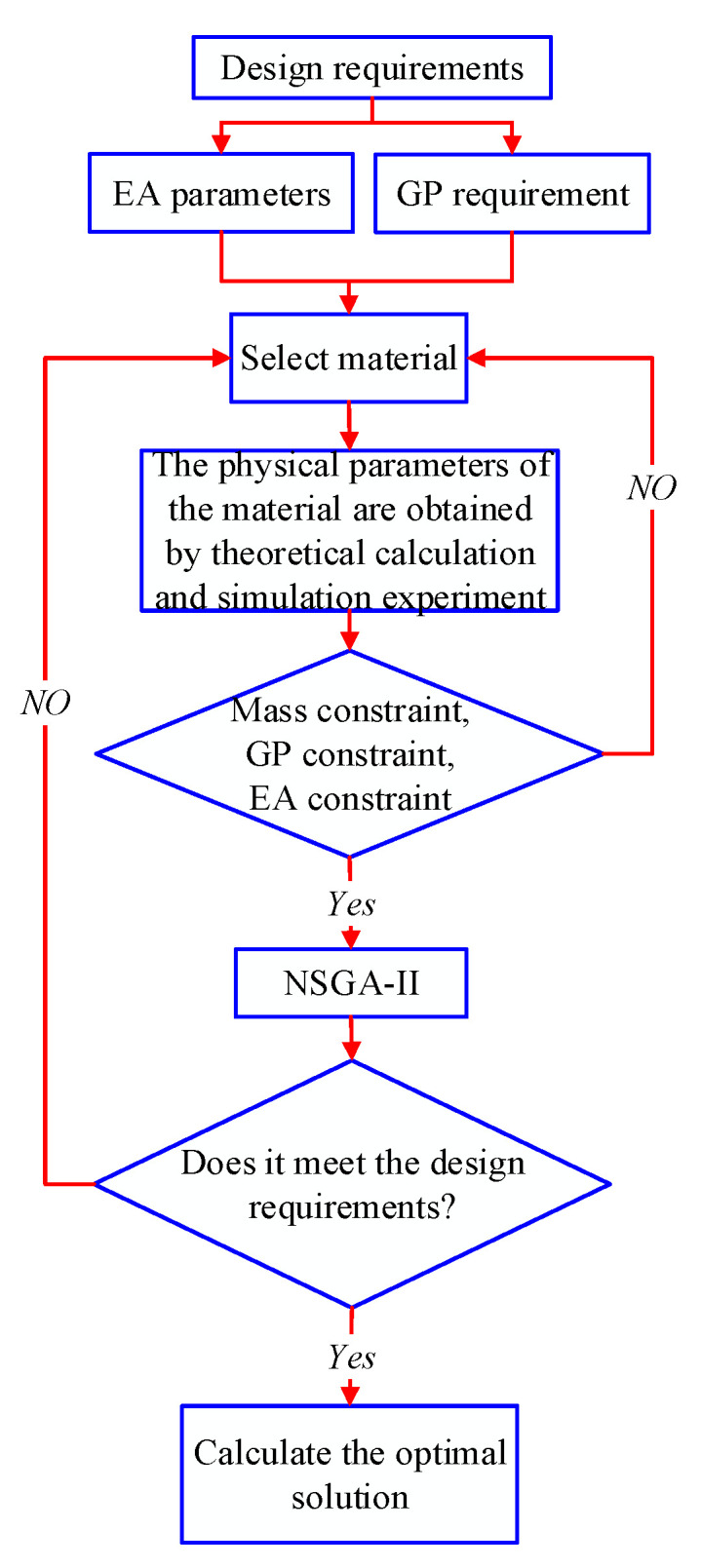
Flowchart of the optimization design.

**Figure 3 materials-13-03340-f003:**
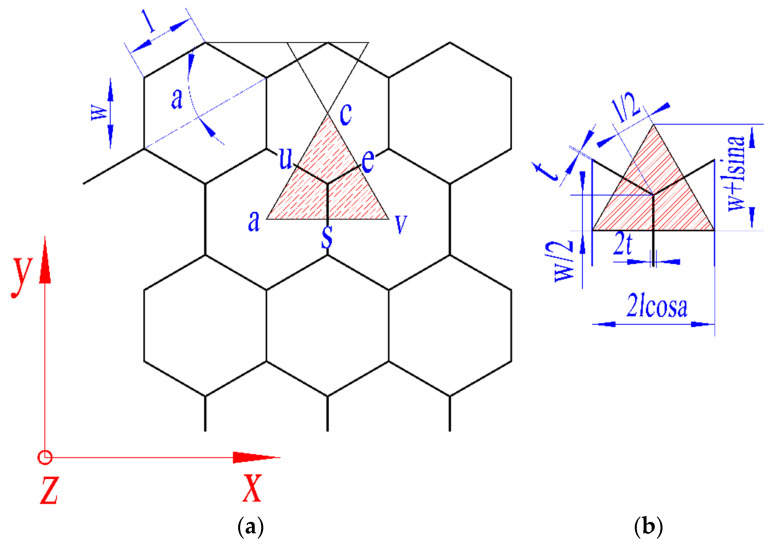
Hexagonal honeycomb structure: (**a**) cross section of hexagonal honeycomb and (**b**) cross section Y-cell.

**Figure 4 materials-13-03340-f004:**
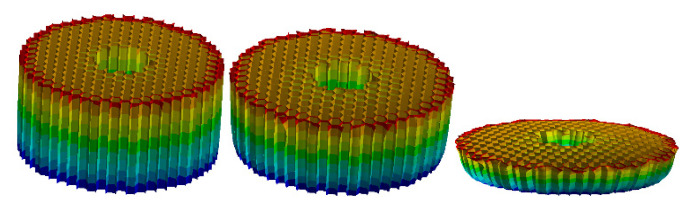
Simulation results of static pressure.

**Figure 5 materials-13-03340-f005:**
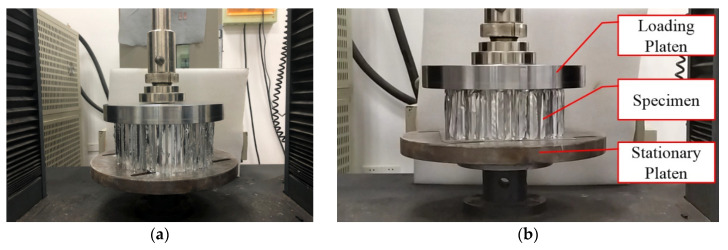
Static pressure experiment of material. (**a**) Placement of experimental material, (**b**)The process of static pressure experiment.

**Figure 6 materials-13-03340-f006:**
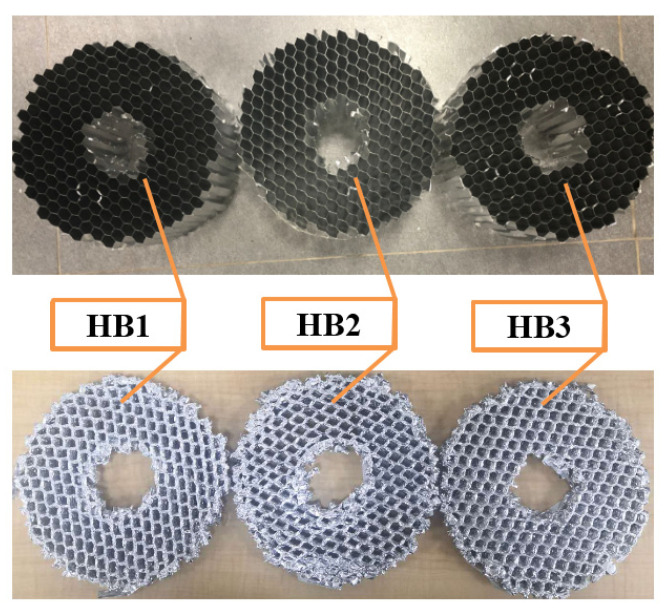
Static pressure experiment results.

**Figure 7 materials-13-03340-f007:**
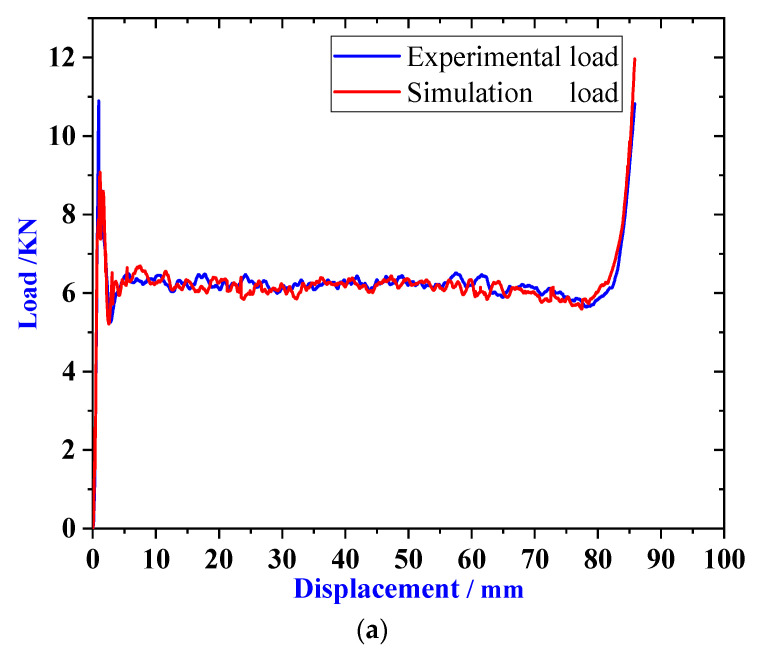
Load-displacement curve of (**a**) HB1, (**b**) HB2, and (**c**) HB3.

**Figure 8 materials-13-03340-f008:**
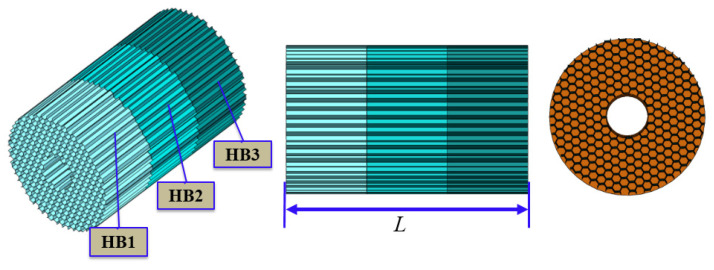
Buffer material collocation method.

**Figure 9 materials-13-03340-f009:**
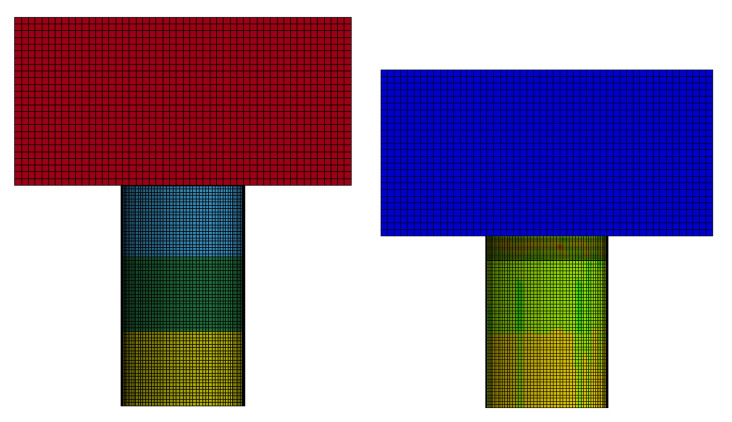
Simulation experiment of series combined impact of honeycomb structure.

**Figure 10 materials-13-03340-f010:**
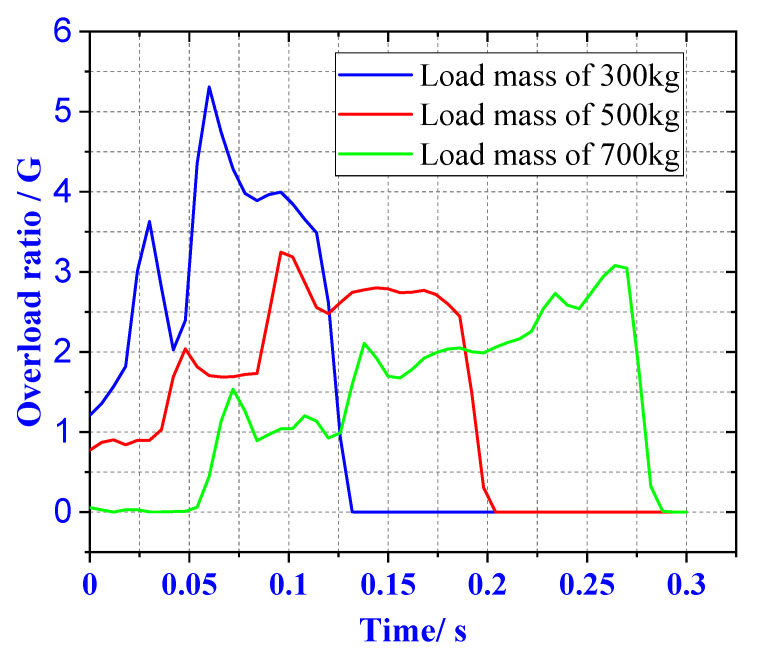
Three load overload rate curves.

**Figure 11 materials-13-03340-f011:**
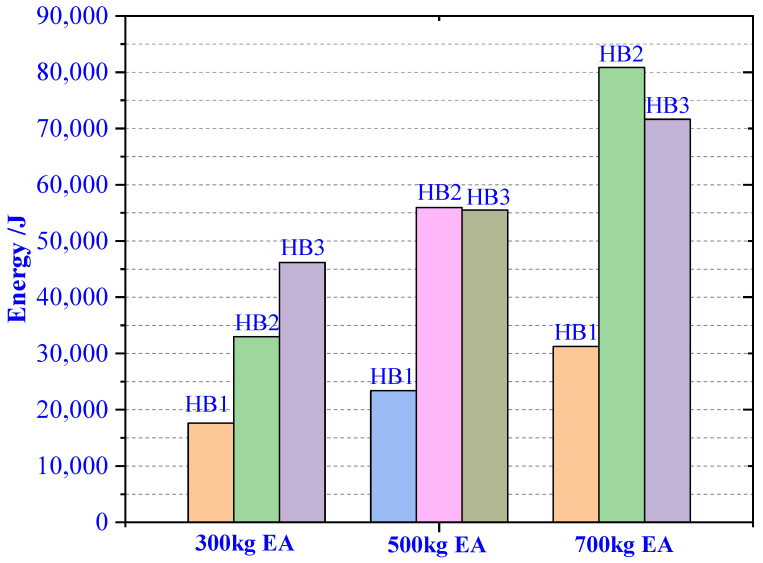
Three masses of material absorb energy.

**Figure 12 materials-13-03340-f012:**
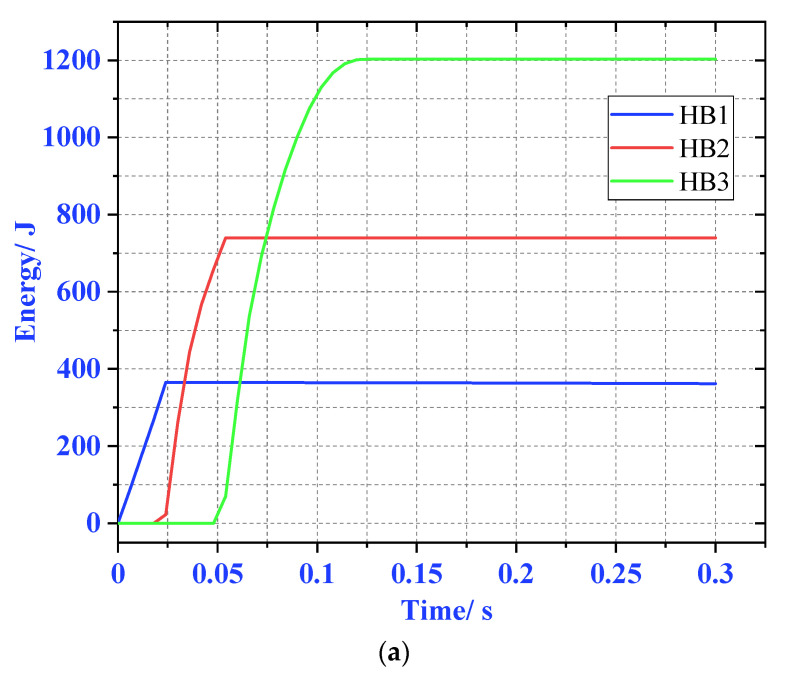
Energy absorption (EA) of three aluminum honeycombs (AHs) under three loads. (**a**) 300 kg, (**b**) 500 kg, and (**c**) 700 kg.

**Figure 13 materials-13-03340-f013:**
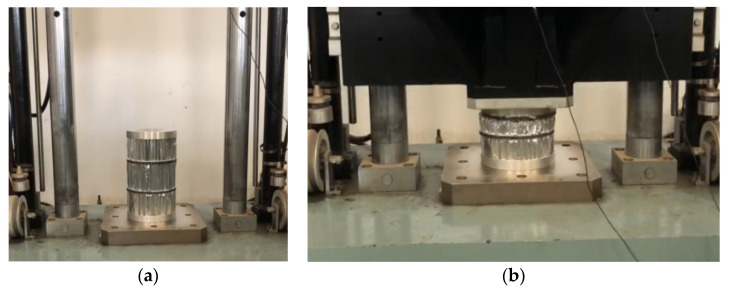
Impact test of AH combination in series. (**a**) Preparation for impact test; (**b**) Impact test completed.

**Figure 14 materials-13-03340-f014:**
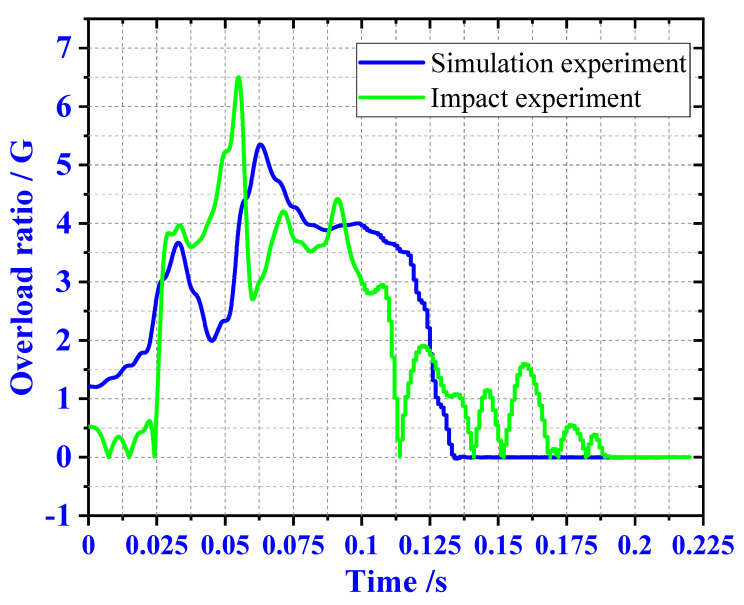
Comparison of experimental results.

**Figure 15 materials-13-03340-f015:**
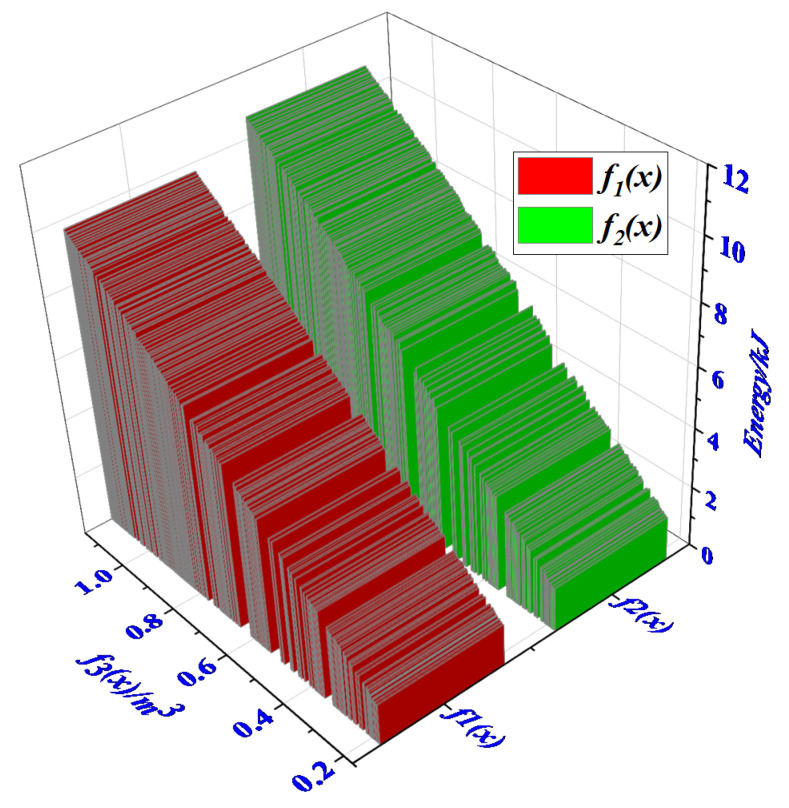
Relation between mass specific energy absorption (MSEA) and volume specific energy absorption (VSEA).

**Figure 16 materials-13-03340-f016:**
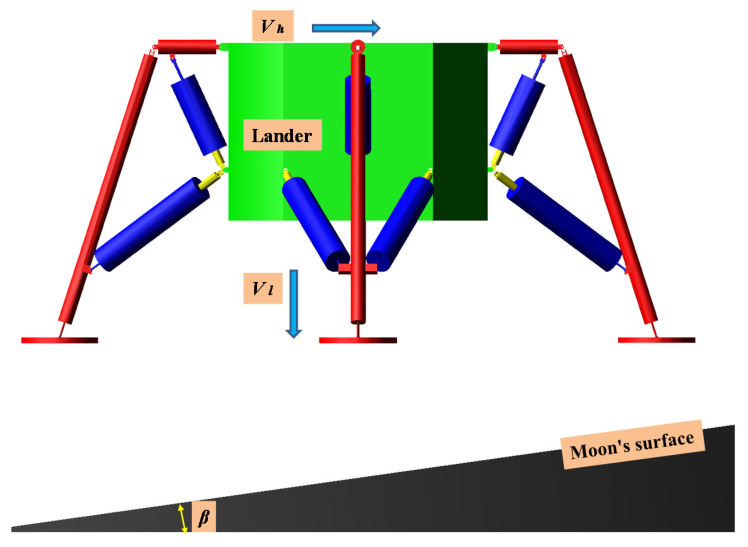
Landing conditions.

**Figure 17 materials-13-03340-f017:**
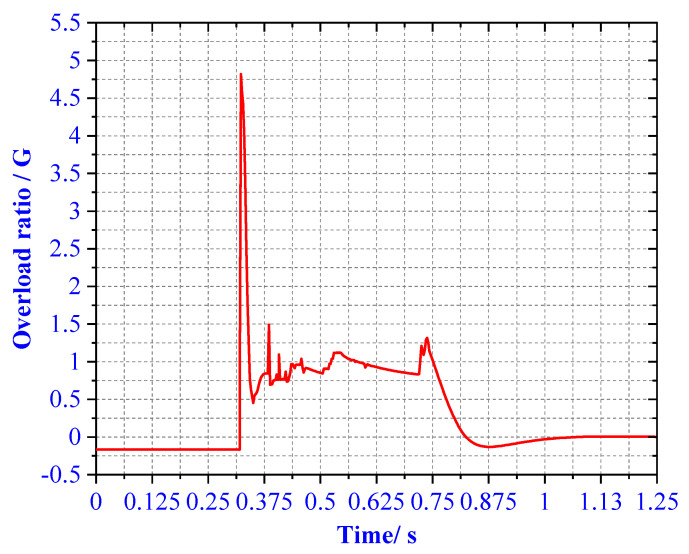
Overload rate of 4 type landing.

**Figure 18 materials-13-03340-f018:**
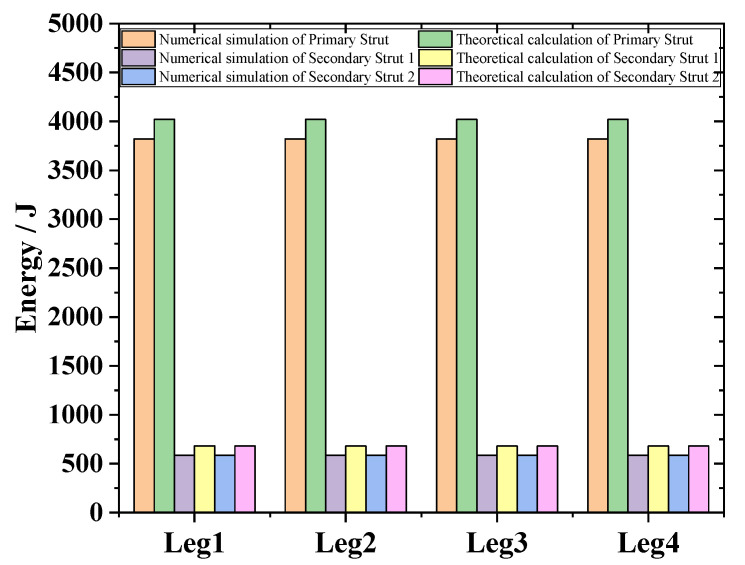
EA of 4 type landing.

**Figure 19 materials-13-03340-f019:**
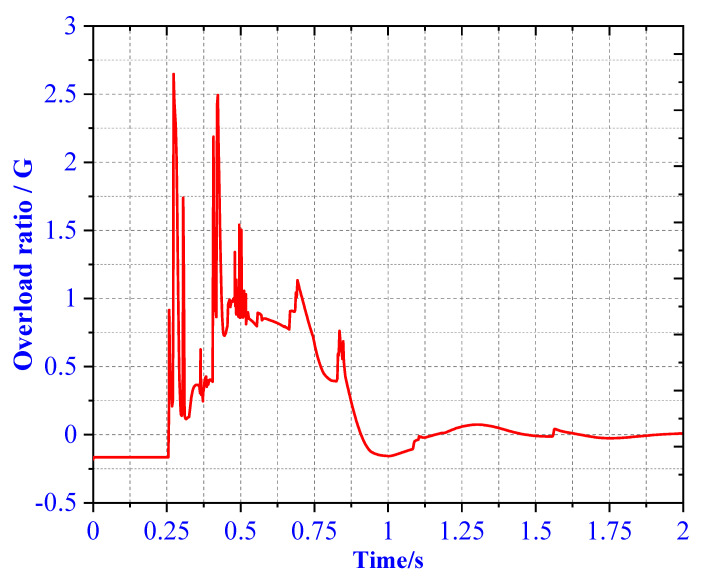
Overload rate of 2-2 type landing.

**Figure 20 materials-13-03340-f020:**
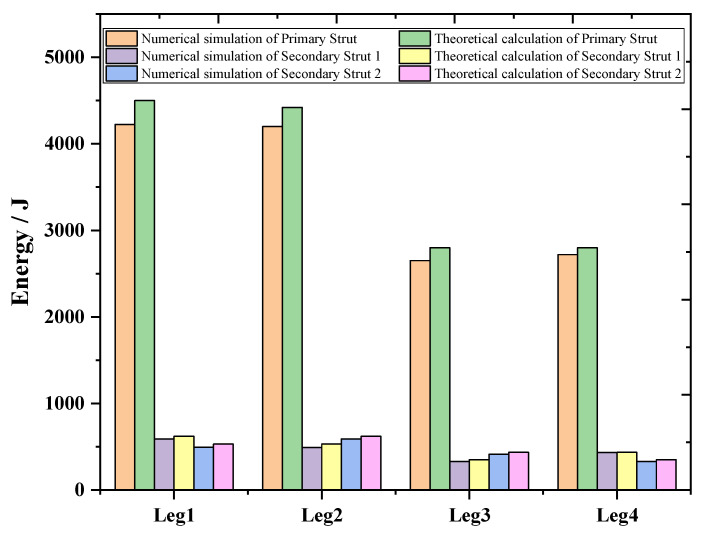
EA of 2-2 type landing.

**Figure 21 materials-13-03340-f021:**
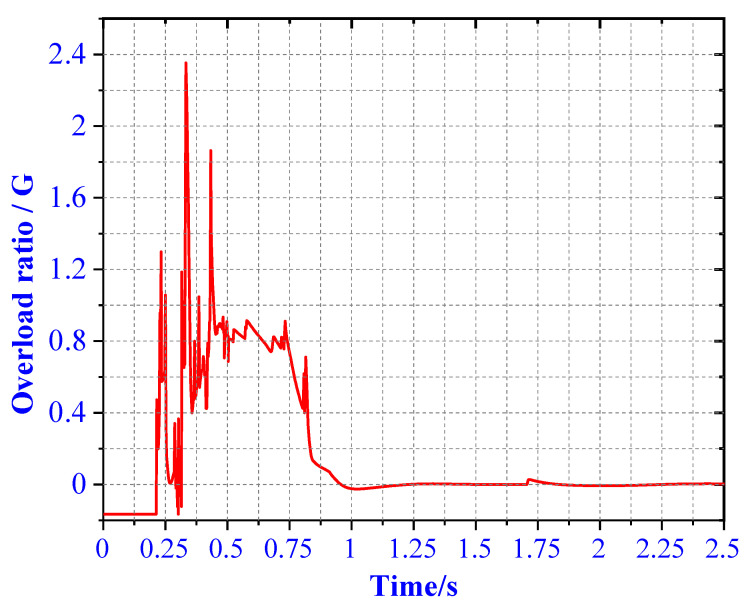
Overload rate of 1-2-1 type landing.

**Figure 22 materials-13-03340-f022:**
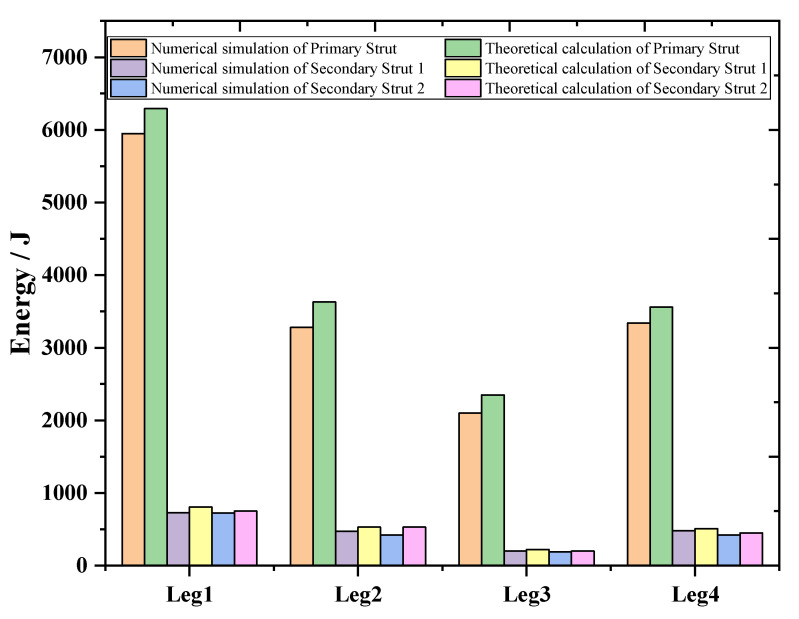
EA of 1-2-1 type landing.

**Table 1 materials-13-03340-t001:** Buffer material experiment parameters.

Parameter Number	HB1	HB2	HB3
Materials	3003	3003	5052
Density/(kg/m^3^)	30	36	50
Initial yield strength/(MPa)	130	135	135
Elasticity modulus/(GPa)	40.39	41.02	41.73
Poisson’s ratio	0.33	0.33	0.33
Thickness of Aluminum Foil/(mm)	0.04	0.05	0.05
Y-cell length/(mm)	6	6	6
Outer diameter/(mm)	190	190	190
Inner diameter/(mm)	50	50	50
Height/(mm)	100	100	100

**Table 2 materials-13-03340-t002:** Experiment compression ratio.

Materials	Initial Simulation Length/(mm)	Initial Experiment Length/(mm)	Experimental Compression Length/(mm)	Simulated Compression Length/(mm)	Experimental Compression Ratio/(%)	Simulation Compression Ratio/(%)
HB1	100	100	15.52	15.05	84.95	84.8
HB2	100	100	16.44	15.96	84.04	83.56
HB3	100	100	18.18	17.87	82.13	81.82

**Table 3 materials-13-03340-t003:** Experimental parameters of cushioning material stamping simulation.

Load Mass/(kg)	HB1/(mm)	HB2/(mm)	HB3/(mm)	Total Length/(mm)	Longitudinal Impact Speed/(m/s)
300	100	100	100	300	4
500	150	150	150	450	4
700	200	200	200	600	4

**Table 4 materials-13-03340-t004:** Dynamic EA results of regular hexagon honeycomb structure.

Materials	EA of 1 kg Honeycomb Material $\left(SEA_{m}\right)$/(KJ)		EA of 1 $m^{3}$ Honeycomb Material $\left(SEA_{v}\right)$/(KJ)	
	Theoretical Value	Simulation Value	Theoretical Value	Simulation Value
HB1	7.901	8.301	219	249
HB2	9.139	9.102	321	345
HB3	10.17	9.941	439	458

**Table 5 materials-13-03340-t005:** Property parameters of stages and material.

Components	Material	Density/(kg/m^3^)	Elastic Modulus/(MPa)	Yield Stress/(MPa)	Poisson Ratio
Constraint structure	Steel	7850	2.1e5	2.05e2	0.3
Struts tubes	Al	2850	7.0e4	5.5e2	0.3
swinging beam	Al	2850	7.0e4	5.5e2	0.3
Landing surface	WOOL	1300	1.2e3	50	0.3
Main leg	Al	2850	7.0e4	5.5e2	0.3
Main body	Steel	7850	2.1e5	2.05e2	0.3

**Table 6 materials-13-03340-t006:** Structure parameter of the ML.

Category	Initial Length/(mm)	Maximum Diameter/(mm)
Primary strut	1500	225
Secondary strut-1	1500	180
Secondary strut-2	1500	180
Swinging beam	500	180
Main leg	2600	180

**Table 7 materials-13-03340-t007:** Coulomb friction parameter defined in contacts of ML.

Contact Name	Decay Constant	Mu Static	Mu Dynamic
Struts vs. Struts	0.85	0.12	0.11
Landing surface vs. footpad	0.85	0.40	0.6

**Table 8 materials-13-03340-t008:** Load case property and initial condition of ML simulation.

Category		Horizontal Velocity at Touch $v_{h}$/(m/s)	Longitudinal Velocity at Touch $v_{l}$/(m/s)	Gravity Value/ $\left(m/s^{2}\right)$	Slope Angle of Landing Surface β/(°)	Load Value/(kg)
**Case 1**	4 legs landing	0	4.0	1.633	0	1200
**Case 2**	2-2 type landing	1.0	4.0	1.633	8	1200
**Case 3**	1-2-1 type landing	1.0	4.0	1.633	8	1200

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
