# Peer review of "Improving the Buffer Energy Absorption Characteristics of Movable Lander-Numerical and Experimental Studies"

_materials, 2020, doi:10.3390/ma13153340_

Round 1
Reviewer 1 Report
The paper presents a study of optimisation of a system of landing energy absorption for a movable lander. This topic could be interesting for a reader of the journal, however, the presented paper is not focused enough on characterisation, modelling and experimental investigation of honeycomb materials applied in the design. The second drawback lays in the lack of clear presentation of the methodology, including the overall description of the landing system, formulation of the optimisation problem, numerical modelling, and experimental research. In my opinion, the paper can be published after major revision.
Detailed comments :
1. In the literature review(Introduction) the Authors cite only selected, recent publications, but omit the fundamental work done in 1960-1980thies.
It has been corrected in the article.
Since the 1960s, research on the buffer characteristics of the lunar lander during landing has been widely carried out. A comparative study of the stability of the three-legged lander and the four-legged lander by Lavender found that the envelope diameter of the three-legged lander was slightly larger than that of the four-legged lander, but the three-legged lander had a smaller overall weight [26]. Robert and Warner et al. proposed a method to solve the elastic rebound problem of the lander by improving the buffer materials [27]. The theoretical model and the results of a parametric study are given in terms of ground slope (5°,7°,??? 15°)[28]. Blanchard carried out impact simulation test on the full-size lunar lander model under lunar gravity with the simulation analysis method, and obtained good test results [29]. Hilderman.et al. verified the cushioning reliability of landing by studying the EA characteristics of the landing buffer under various harsh landing conditions [30]. William introduced the development of lunar module landing gear subsystem in the Apollo 11 lunar landing mission in detail, and verified the lander test analysis method under different working conditions [31].
In addition, the analysis of landing overload performance has also attracted great attention from the literature. Jones and Hinchey [32] and Black [33] have studied the impact of landing environment on airframe overload response, providing guidance for the study of buffering characteristics. In order to study the impact of different influencing factors on the overload response of the Lunar lander, Alderson and Wells [34] created mathematical models and computer programs. Reference [35] conducted a landing experiment on the full-size lander and verified the landing reliability.
[26]. Lavender R E. Monte carlo approach to touchdown dynamics for soft lunar landing NASA TN D23117. US: ANSA, 1965.1-52.
[27]. Robert W. Warner, Donald R. Marble. Possible Materials Needs for Energy Absorption in Space-vehicle Landings, NASA-TM-X-54070,1964.
[28]. BLACK, R. J. Quadrupedal landing gear systems for spacecraft. 1964.196-203.
[29]. Blanchard UJ. Full-scale dynamic landing-impact investigation of a prototype lunar module landing gear. NASA Technical Report NASA-TN-D-5029, Hampton, VA, United States: NASA Langley Research Center, 1 March 1969.
[30]. Hilderman, R. A.; Mantus, M.; Mueller, W. H. Landing dynamics of the Lunar Excursion Module. Journal of Spacecraft and Rockets, 1966, 3(10):1484-1489.
[31]. William, F.; Rogers. Apollo experience report-lunar module landing gear subsystem, NASA- TN-D-6850, 1972. 1-45
2. In my opinion, the section "Introduction to ML" must be extended. The landing process and mechanism of walking of the movable lander should be described. The Authors present a functional description of the. Figure 1 presents the inner structure of a secondary strut, which is not further explained in detail in the text.
This information is crucial for the reader to understand requirements, proposed mechanisms of energy absorption and constraints imposed on the system.
It has been corrected in the article.
The structure of primary strut and secondary strut is composed of inter tube, outer tube, aluminium honeycomb-I, aluminium honeycomb-II, driving mechanism, ball screw, screw nut, and push-pull locking mechanism. The inter tube of the primary strut is connected with the rotary joint, and the inter tube of the secondary strut is connected with the spherical joint, which is also the only difference between the primary strut and secondary strut.The primary strut and the swinging beam are connected by a rotating joint. The primary strut and the swinging beam are connected to the body through the rotary joint. The joints of the main leg connecting the swinging beam and the foot pad are a universal joint and a spherical joint, respectively. The two secondary struts and the main leg are connected by the ball joint. The main body and the auxiliary pillar are connected by a universal joint. The primary struts are crushed by the internal AH to buffer the vertical impact of the main body. The auxiliary pillar needs to withstand the vertical and lateral swing restraint forces exerted by the main leg, and the restraint force is achieved by the compression EA of the AH material inside the auxiliary pillar.
Working principle of the ML:
In the initial state, the buffer mechanism and the drive mechanism are independent of each other, the inter tube is locked with the push-pull locking mechanism, and the push-pull locking mechanism is installed in the middle of the aluminum honeycomb-I and aluminum honeycomb-II. At the same time, the screw nut is located on the left side of the ball screw. As shown in Figure 1. No connection has been established between the lead screw nut and the inner tube. During the soft landing on the lunar surface, due to the impact of different landing speeds and irregular lunar surface environment, the stress direction of the four landing legs and landing inclination Angle of the ML have many uncertainties. After landing, the main leg and swinging beam transmit the contact force between the foot pad and the lunar surface to the primary struts and secondary struts s through the inter tube. Because the direction of the force is different, each primary strut and secondary strut are subjected to tension or pressure, and the aluminum honeycomb I or aluminum honeycomb II is compressed by the push-pull locking mechanism to achieve cushioning.
After the soft-landing mission is completed, the ML needs to complete attitude adjustment or walking preparation work. At this time, the driving mechanism starts to move and drives the ball screw to rotate. Because the circumferential freedom of the screw nut is limited, the screw nut can only move horizontally. When the screw nut moves to the locking position of the inter tube, it is locked together with the inter tube, and the lock between the inter tube and the push-pull locking mechanism is released. The screw nut drives the inner tube of the primary struts and secondary struts to drive the swing beam and the main leg to swing, and finally achieves walking.
3. Section 3 "Algorithm" does not provide enough information about the proposed algorithm. Flowchart of the optimisation (Fig.2) raises a lot of questions and is not commented in detail in the text. Authors must explain what is the NSGA-II algorithm, and how it is related to the reference [31] (line 129). The Authors formulate only the very general multiobjective Pareto optimisation problem.
It has been corrected in the article.
The fast and elitist non-dominated sorting in genetic algorithms (NSGA-II) is an advanced multi-objective optimization method based on evolutionary algorithm EA. This algorithm not only reduces the complexity of non-inferior sorting genetic algorithm, but also has the advantages of faster running speed and better convergence of solution set. NSGA-II can find Pareto optimal solution set that makes each objective function as large as possible (or as small as possible), and thus provides an effective tool for balancing between each objective function.
In Figure 2, We optimized the EA parameters and GP (such as material length, cross-sectional area, etc.) of the buffer material according to the requirements of the ML's buffer EA. Firstly, we will get the EA parameters and GP requirement of the buffer material through theoretical calculation and simulation. If the material parameters meet the material's mass constraint, the GP constraint and EA constraint. Next, we will take the minimum total mass of
Swinging beam Foot padPrimary strutMain legSecondary strut-2Secondary strut-1Main bodyPush-pull locking mechanismDrivingmechanismIntertubeAluminumhoneycomb-IIAluminumhoneycomb-IBall screwOuter tubeRotary jointSpherical jointscrew nut
the material, the VEA and MEA as the objective function. Finally, the NSGA-II is used to obtain the optimization results that meet the design requirements.
4. Line 165, reference [32], which is the basis for the analysis of the honeycomb structure is a very short, 3-page long ESA report and does not contain any reference shock-absorbing structures. Authors should at least briefly comment on their method of analysis of the honeycomb, eq.2 should be explained.
It has been corrected in the article.
In order to obtain the compression performance of the three honeycomb materials under out-of-plane static compression or impact load, based on the symmetric structure of the honeycomb structures and law of energy conservation. Based on this, average compressive stress constitutive equation under out-of-plane compression is deduced with Mises yield criterion and Tresca yield criterion. Considering the effect of strain rate on its mechanical characteristics based on the Cowper-Symonds model, dynamic average compressive stress calculating equation is deduced. In order to reduce the influence of the honeycomb structure of the same specification due to different external dimensions on the average force, in this paper, the average stress of the honeycomb structure is used instead of the average load. Using the method in [44,45], calculate the percentage of Y-cell in Fig.2(b). The area of Y-Cell can be obtained by Eq. (1).
??=??=??0?2(?+2?)?√??(?+2????(?+?????)) (2)
Where, ?0 denotes the initial yield strength of the honeycomb matrix material; ? denotes the load applied to the surface Y-Cell. ? denotes the area of Y-Cell. ? denotes the compression ratio, that is the effective compression length coefficient, as shown in Table 2.
5. The adopted method is based on RVE approach, and the unit cell will not describe well the material at the boundary. In case of a small number of cells in the structure this could influence the results - could Authors comment on this?
It has been corrected in the article.
The purpose of the division of small cell structure is to facilitate the calculation of the relationship between cell volume and energy absorption characteristics. The equivalent volume element (RVE) method solves the continuity problem of model elements in finite element analysis. At the same time, it avoids the complicated calculation of using multiple continuous element models, which makes the calculation of energy absorption of the model simpler.
6. Line 184, please correct or explain loading conditions for the quasi-static test of the honeycomb (50kN, 5mm/min). Is it a displacement or load driven test?
It has been corrected in the article.
The testing machine adopts displacement driving test, the loading force is 50?? and the speed is 5??/???.
7. Experimental data should be more extensively reported (stress-strain or load-displacement curves) - e.g. in Fig.7 instead of general crushing characteristics.
It has been corrected in the article.
8. What is the correlation of the experiment with average stress calculated from eq.2?
It has been corrected in the article.
The honeycomb structures with different geometric parameters have different average stresses, and the accuracy of the pressure load is verified by calculating the average stresses, so as to study the energy absorption characteristics of hexagonal honeycomb structures.
9. Simulation of the experiment is not described and commented at all. Please additionally compare the experimental load curve with the numerical results.
It has been corrected in the article.
Under the condition of static pressure load, simulation experiment and static pressure test were carried out on the buffer material, and the load-displacement curves of HB1, HB2 and HB3 were obtained, as shown in figure 7(a)-(c). It can be observed that the simulation results agree with the experimental results well. The difference between the experimental and numerical curves may be resulted from the rupture of the material in the experimental test while the numerical models did not take into account the failure of material. Through experiments we have obtained the average loads of HB1, HB2 and HB3 are 6.25??, 8.27?? and 9.55??, respectively.
10. Why section 4.2 is entitled "Dynamic EA...". The proposed model is fully quasi-static.
It has been corrected in the article. 4.2 EA characteristics
11. Please explain the origin of eq.6; why eq. 5 and 7 are directly dependent on the system of units?
It has been corrected in the article.
??=20−3?0??[ 1+(?08?1?√√36??(?+2?))1?] (√√36??(?+2?)−?1?)? (6)
Where, ?=30?, ?=?, ? denotes the height of the Y-cell; ? denotes the Strain rate sensitivity coefficient of material; ?0 denotes the Initial velocity of impact; ? denotes the Strain rate sensitivity coefficient of material; ?1 denotes the height compensation factor. Based on the difference between the effective compression height and the theoretical compression height is calculated based on simulation and experiment,choose between 1.4~1.6,we choose ?1=1.6[46].
??=10−3??????? (5)
Where, ??? denotes the dynamic mean stress of honeycomb structure under impact load, ?? denotes the ultimate strain of the Y-cell; ? denotes the cross-sectional area of the Y-cell. and ? denotes the height of the Y-cell of the honeycomb.
{?=10−9?ℎ?=10−9(??+??)????=10−9??=10−9?????(?+?????)? ((7))
?? denotes the base material density, corresponds to the actual material density of the aluminum honeycomb material,,?ℎ denotes the material structure density,,the density of solid material of the same mass and volume as aluminum honeycomb material. S denotes cross section area of Y-cell.
12. Please extensively comment eq. 6 and 7. What is k and k1, why k1=1.6 was chosen?
It has been corrected in the article.
? denotes the compression ratio, that is the effective compression length coefficient, as shown in Table 2.
?1 denotes the height compensation factor. Based on the difference between the effective compression height and the theoretical compression height is calculated based on simulation and experiment,choose between 1.4~1.6,we choose ?1=1.6[46].
13. The methodology of the dynamic simulation in section 4.3 is not described at all (method, software, mesh, BC, material models, etc.)
It has been corrected in the article.
In this paper, ANSYS software is used as the pre-processing software for finite element analysis, and LS-DYNA software is used to solve the impact of aluminum honeycomb materials. In order to improve the computing speed, the grid cell size of HB1, HB2 and HB3 was set as 0.5mm. Honeycomb of Nonlinear- Foam Material Models is used as a Material model in HB1, HB2 and HB3. Material parameters are shown in Table 1. As shown in Figure. 9, the buffer material model was placed between the rigid mass block and the fixed surface, and the bottom HB3 material was fixed. The reference point method [47] is adopted to determine the boundary conditions and initial velocity of the impact mass block. All degrees of freedom are limited except for translation along the direction of collision. When three material models in series are impacted by a mass block, in order to prevent the contact between each surface of the model from penetrating, the contact type was selected to be single-side automatic contact. In order to ensure that the simulation results of metal honeycomb impact are close to the real results, the dynamic friction coefficient between the mass block and the aluminum honeycomb material is set as 0.2.
[47]Riccio A , Raimondo A , Saputo S , et al. A numerical study on the impact behavior of natural fibers made honeycomb cores[J]. Composite Structures, 2018, 202(OCT.):909-916.
14. What is "overload ratio response"? Units show that it's acceleration/deceleration. This should be corrected further in the text.
It has been corrected in the article.
The ratio of the lander's opposite acceleration to the earth's gravitational acceleration (G) is called the overload rate. ?=9.8?/?2. During the soft landing of lunar lander, the maximum peak acceleration has a great influence on the safety of precision instruments and astronauts. In the aerospace field, the overload rate of the lander has been used as a key index to evaluate its safety performance.
15. Authors should explain, why the maximum peak acceleration was chosen as the key objective in the design. Different measures can be applied based on application requirements.
It has been corrected in the article.
During the soft landing of lunar lander, the maximum peak acceleration has a great influence on the safety of precision instruments and astronauts. In the space field, the maximum peak acceleration of the lander has been used as a key index to evaluate its safety performance.
16. Line 276 - is the mass maximised during optimisation?
It has been corrected in the article.
Yes,the mass is maximised during optimization.
17. The optimisation problem is not formulated in detail. What are the variables of the problem?
It has been corrected in the article.
When optimizing the EA characteristics of the aluminium honeycomb structure of the ML, we take the maximum VSEA and maximum MSEA of the three materials as optimization objectives, the mass and volume of aluminium honeycomb materials as design variables. The optimization goal of this paper is to reduce the overall mass and volume of the buffer material as much as possible. In order to further improve the EA characteristics of the buffer material. As the outer and inner diameters of the buffer materials are restricted by the structural parameters of the landing leg and the internal drive system, The three target functions of maximum VSEA ?1(?), maximum MSEA ?2(?), and maximum mass ?2(?) of the three materials need to be evaluated.
18. In section 5, Authors refer to the NSGA-II algorithm, which is not described in detail. Most probably it's a genetic or evolutionary optimisation method.
It has been corrected in the article.
The NSGA-II is an advanced multi-objective optimization method based on EA. This algorithm not only reduces the complexity of non-inferior sorting genetic algorithm, but also has the advantages of faster running speed and better convergence of solution set. The NSGA-II can obtain the optimal frontier of multi-objective optimization model efficiently by the improved fast non-dominated ranking and elite selection strategy without any parameter adjustment.
19. The language is generally on an acceptable level, however, it requires correction. In some cases, sentences do not have a clear meaning (e.g. line 303, line 324, line 334, etc.). Spelling should be checked (e.g. Table 5).
It has been corrected in the article.
20. Quality of figures must be improved.
It has been corrected in the article.
Figures 1, 3, 4,5, 6, 7, 10, 11, 12, 14, 15, 16, 17, 18, 19, 20, 21 and 22 have been replaced with higher quality versions.
Reviewer 2 Report
The manuscript proposes a design procedure of a new-type composite energy-absorbing structure by combining the characteristics of AH structures for application as a buffer in movable landers. The topic of the paper is practical and fits the aims and scope of the journal. Moreover, the paper is very well structured and the results are clearly analyzed. The authors have also compared their results with those of numerical simulation to confirm the validity of the proposed approach. The conclusion section covers the achievements of the paper very well.
Consequently, I would recommend the publication of the paper after the following minor rooms will be addressed by the authors:
1. In the literature review section, the gap analysis should be more highlighted to justify the necessity of doing this piece of research.
It has been corrected in the article.
At present, many kinds of buffer structures have been widely used in industry and aerospace, such as airbag buffer [9], spring damping buffer [10], hydraulic buffer [11] and compressible material buffer [12]. During the study of the cushioning characteristics of the ML, both the airbag buffer and the spring damping buffer are not conducive to the control and are easy to cause the spacecraft to bounce and roll. The buffer fluid in the hydraulic buffer structure has been leaking in the past, which has limited its application in planetary exploration. Therefore, the compressible buffering method with simple structure, small mass, and not easy to bounce or roll is taken as the research object.
The lightweight and reliability of the lander has become the key technology to optimize the cushion performance. The AH structures are of demonstrative advantages of lightweight, excellent load bearing capacity and EA efficiency, which have been extensively used as energy absorbers in automotive, railway, naval and aerospace industries. The experimental methods [13,14] analytical methods [15] and numerical methods [16,17] are generally used to test mechanical properties of honeycomb materials. Due to the good crushing effect of honeycomb on the outside, a large number of experimental studies have been conducted on its dynamic compressive strength, and it is found that the thickness [18],side length [19,20] impact velocity [21] has the most obvious influence on the EA characteristics of the material.
The combination of experimental and Digital Image Correlation results worked excellently well in the determination of global and local deformation mechanism of the honeycomb core Khan [22]. A compression/shear coupling with transverse shear is introduced to predict the response of honeycomb sandwich structures to impact loads [23]. The dynamic impact crushing behavior of multi-layer honeycomb sandwich panels and the impact ensile loading behavior of their material members are examined in this experimental investigation [24].
The main modeling methods for the EA of buffer materials are finite element method(FEM)and multi-body system dynamics method. Among them, the FEM can consider the nonlinear characteristics of geometric materials and contacts, so it can usually get more accurate results. One disadvantage of this method is that it is too time-consuming to analyze the multi-body dynamic model. Although the accuracy is not high, it can greatly improve the computational efficiency. In this paper, the physical parameters of the buffer material were obtained by using the finite element method. The multi-body system dynamics method was selected for the study of the soft-landing characteristics of the whole machine, and was compared with the theoretical calculation results [25].
Since the 1960s, research on the buffer characteristics of the lunar lander during landing has been widely carried out. A comparative study of the stability of the three-legged lander and the four-legged lander by Lavender found that the envelope diameter of the three-legged lander was slightly larger than that of the four-legged lander, but the three-legged lander had a smaller overall weight [26]. Robert and Warner et al. proposed a method to solve the elastic rebound problem of the lander by improving the buffer materials [27]. The theoretical model and the results of a parametric study are given in terms of ground slope ( [28]. Blanchard carried out impact simulation test on the full-size lunar lander model under lunar gravity with the simulation analysis method, and obtained good test results [29]. Hilderman et al. verified the cushioning reliability of landing by studying the EA characteristics of the landing buffer under various harsh landing conditions [30]. William introduced the development of lunar module landing gear subsystem in the Apollo 11 lunar landing mission in detail, and verified the lander test analysis method under different working conditions [31].
Lunar landers around the world need a lot of experiments before they can be launched,There are too many unknown difficulties in the real prototype experiment due to its shortcomings such as long manufacturing cycle, difficult data acquisition and large assembly error. The numerical simulation technology is widely used in the aerospace field,due to its brand-new research and development mode, which has the characteristics of low research and development cost, short research and development cycle and high product quality. Due to the soft-landing characteristics and overload rate of the ML, its safety is greatly affected. At present, there is only research on the soft-landing characteristics of the traditional lunar lander, and there is no research data on the buffering characteristics of the ML.
In addition, the analysis of landing overload performance has also attracted great attention from the literature. Jones and Hinchey [32] and Black [33] have studied the impact of landing environment on airframe overload response, providing guidance for the study of buffering characteristics. In order to study the impact of different influencing factors on the overload response of the Lunar lander, Alderson and Wells [34] created mathematical models and computer programs. Reference [35] conducted a landing experiment on the full-size lander and verified the landing reliability.
The reliability of soft landing is a very important technical index in the process of lander development. For example, the design process of Apollo lander (USA) [36]. Research objectives of lander mainly focus on the EA characteristics and conceptual design of soft landing [37]. In addition, methods to study the cushion characteristics of the lander can be generally divided into three categories [38], namely theoretical analysis method [39], simulation analysis method [40] and experimental method. In particular, simulation method is commonly used at present, but it needs to find the optimal buffer material collocation through repeated iteration and simulation for many times. Therefore, in order to obtain the best geometric parameters (GP) of buffer materials, it is necessary to combine mathematical calculation method to optimize the combination and collocation mode of buffer materials and quickly find the optimization method of the EA on the basis of theoretical research.
From previous studies, it can be seen that the EA of the AH and other metal buffer materials depend on their GP. In order to estimate the EA of the AH, the load mass of the lander, the lateral velocity, horizontal velocity and landing environment should be considered. On this basis, the influence of cushion material on the overload coefficient of the body under the impact load of the ML is studied. In this paper, a more effective method using theoretical derivation, simulation experiment and NSGA-II optimization method for landing EA analysis is proposed.
This article will provide theoretical basis and technical support for the design of the ML’s buffer structure based on the theoretical analysis and numerical simulation, and by optimizing the design of the EA characteristics of the aluminum honeycomb. Numerical simulation and virtual prototype technology are used to design a reliable EA buffer structure with lower cost in a short development cycle. This research result will play a certain role in promoting the development of cushion technology for mobile lander, at the same time, it can also be extended to other research fields of buffer EA.
2. Figures 1,3,7,10,11, and 12 should be replaced with higher quality/resolution versions.
It has been corrected in the article.
Figures 1-22 have been replaced with higher quality/resolution versions.
3. Figure 2 is the main contribution of the paper. Please add more analytic discussion about it.
It has been corrected in the article.
In Figure 2, We optimized the EA parameters and GP (such as material length, cross-sectional area, etc.) of the buffer material according to the requirements of the ML's buffer EA. Firstly, we will get the EA parameters and GP requirement of the buffer material through theoretical calculation and simulation. If the material parameters meet the material's mass constraint, the GP constraint and EA constraint. Next, we will take the minimum total mass of the material, the VEA and MEA as the objective function. Finally, the NSGA-II is used to obtain the optimization results that meet the design requirements. This optimization method provides a new research direction for the research of the cushioning EA of the ML: the user can calculate the optimal combination of the physical parameters of the buffering materials and their EA changes within the constraints.
Reviewer 3 Report
Investigation on energy absorption structure under impact loads using movable lander was done in the present article which could be modified in different ways.
1.First of all, authors should focus on the application of the buffer and present structures and mention it in introduction section.
It has been corrected in the article.
The structure of primary strut and secondary strut is composed of inter tube, outer tube, aluminium honeycomb-I, aluminium honeycomb-II, driving mechanism, ball screw, screw nut, and push-pull locking mechanism. The inter tube of the primary strut is connected with the rotary joint, and the inter tube of the secondary strut is connected with the spherical joint, which is also the only difference between the primary strut and secondary strut.The primary strut and the swinging beam are connected by a rotating joint. The primary strut and the swinging beam are connected to the body through the rotary joint. The joints of the main leg connecting the swinging beam and the foot pad are a universal joint and a spherical joint, respectively. The two secondary struts and the main leg are connected by the ball joint. The main body and the auxiliary pillar are connected by a universal joint. The primary struts are crushed by the internal AH to buffer the vertical impact of the main body. The auxiliary pillar needs to withstand the vertical and lateral swing restraint forces exerted by the main leg, and the restraint force is achieved by the compression EA of the AH material inside the auxiliary pillar.
Working principle of the ML:
In the initial state, the buffer mechanism and the drive mechanism are independent of each other, the inter tube is locked with the push-pull locking mechanism, and the push-pull locking mechanism is installed in the middle of the aluminum honeycomb-I and aluminum honeycomb-II. At the same time, the screw nut is located on the left side of the ball screw. As shown in Figure 1. No connection has been established between the lead screw nut and the inner tube. During the soft landing on the lunar surface, due to the impact of different landing speeds and irregular lunar surface environment, the stress direction of the four landing legs and landing inclination Angle of the ML have many uncertainties. After landing, the main leg and swinging beam transmit the contact force between the foot pad and the lunar surface to the primary struts and secondary struts s through the inter tube. Because the direction of the force is different, each primary strut and secondary strut are subjected to tension or pressure, and the aluminum honeycomb I or aluminum honeycomb II is compressed by the push-pull locking mechanism to achieve cushioning.
After the soft-landing mission is completed, the ML needs to complete attitude adjustment or walking preparation work. At this time, the driving mechanism starts to move and drives the ball screw to rotate. Because the circumferential freedom of the screw nut is limited, the screw nut can only move horizontally. When the screw nut moves to the locking position of the inter tube, it is locked together with the inter tube, and the lock between the inter tube and the push-pull locking mechanism is released. The screw nut drives the inner tube of the primary struts and secondary struts to drive the swing beam and the main leg to swing, and finally achieves walking.
2. Figures 1, 3, 5, 7, 10, 11, 12, 14, 15, 16, 17, 18, 19, 20, 21, 22 must be improved in quality. They are not readable at all.
It has been corrected in the article.
Figures 1-22 have been replaced with higher quality/resolution versions.
3. In table 3, why did you use 300, 500, and 700(kg) for load mass?
It has been corrected in the article.
In Table 3, considering the different landing environment of the lander, the impact force of each landing leg is greatly different. When four legs land simultaneously, the maximum buffer absorption energy of a single landing leg can be equivalent to the impact energy with a load mass of 300kg. During the 2-2 type landing, the maximum buffer absorption energy of a single landing leg can be equivalent to the impact energy with a load mass of 500kg. During 1-2-1 landing, the maximum buffer absorption energy of a single landing leg can be equivalent to the impact absorption energy when the load mass is 700kg.Therefore, we chose these 300, 500 and 700(kg) as the load masses.
4. In figure 11, the amounts of absorbed energy for 500kg are very close to each other. For 300kg, it is increasing and for 700kg, it increases then decreases. Authors should bring scientific reason for this pattern.
It has been corrected in the article.
In Figure. 11, when the load mass is 300kg, the three materials are completely compressed. With the increase in the density of HB1, HB2 and HB3, the energy absorption of HB1, HB2 and HB3 increases continuously. When the load mass is 500kg, the total length of the three buffer materials increased in equal proportion, HB1 and HB2 are all compressed. HB3 material was not completely compressed, resulting in the energy absorbed by HB2 and HB3 being similar. When the load mass was 700kg, HB1 and HB2 are all compressed. HB3 absorbs less energy than HB2 because of the reduced ratio of length to which HB3 is compressed.
5. In figure 14, after Time>0.137, there is a significant difference between experimental results(Simulation and Impact). Why?
It has been corrected in the article.
In Figure 14, we can see that the variation trend of the numerical simulation results and the experimental results of the overload rate curve is similar. In the actual experiment, due to the friction between the load mass and the guide rail, the vibration between the test bed and the ground, and the sensitivity error of the sensor, there is a fluctuation error between the overload rate and the numerical simulation results. However, the experimental results meet the requirements of the impact test.
6. For your work application you need to bring some industrial implementations of buffer to convey your reason of study. You can find one application in the following study:- Effect of the change in auto coupler parameters on in-train longitudinal forces during brake application
It has been corrected in the article.
At present, many kinds of buffer structures have been widely used in industry and aerospace, such as airbag buffer [41], spring damping buffer [42],hydraulic buffer [43] and compressible material buffer [44]. During the study of the cushioning characteristics of the lander, both the airbag buffer and the spring damping buffer are not conducive to the control and are easy to cause the spacecraft to bounce and roll. The buffer fluid in the hydraulic buffer structure has been leaking in the past, which has limited its application in planetary exploration. Therefore, the compressible buffering method with simple structure, small mass, and not easy to bounce or roll is taken as the research object.
[41] Cao P; Hou X; Xue P; et al. Research for a modeling method of mars flexible airbag based on discrete element theory. 2017 2nd International Conference on Advanced Robotics and Mechatronics (ICARM). 2017.
[42] Saeed M.; Reza S. Effects of the change in auto coupler parameters on in-train longitudinal forces during brake application. Mechanics & Industry, 2015, 16.1-13.
[43] Meran; Ahmad P.; Mugan. Design and analysis of a hydraulic-elastic railcar buffer. Proceedings of the Institution of Mechanical Engineers, Part F. Journal of rail and rapid transit. 2018, Vol. 232.1994–2005.
[14] H.S. Kim, New extruded multi-cell aluminum profile for maximum crash energy absorption and weight efficiency, Thin-walled Struct. 40 (2002) 311–327.
Round 2
Reviewer 1 Report
The paper has been improved after the changes introduced by the Authors, however, I still would like to clarify few points, which were not fully addressed in the reply.
1.Point 6: if the test is displacement driven than the force is resulting from specimen characteristics. Is 50kN the maximal force of the testing machine?
It has been corrected in the article.
The testing machine adopts displacement driving test, the maximal force of the testing machine is and the speed is 5mm/min.
2. Point 8: could Authors give a quantitative answer about the difference between experiment and eq.2
It has been corrected in the article.
Dear reviewer, thank you for your valuable comments. I found that the Eq. (2) had no connection with the experiment and deleted it.
3. Point 11: could Authors describe how the eq.6 is derived or give a reference with a short explanation about the origins of the equation
It has been corrected in the article.
In reference [46], The theoretical formula of dynamic mean stress and ultimate strain of hexagonal honeycomb structures under impact load based on Mises strength criterion is derived. As shown in Eqs. (3) and (4):
Where, denotes the initial yield strength of the honeycomb matrix material; denotes the compression ratio, that is the effective compression length coefficient, as shown in Table 2. , , denotes the Strain rate sensitivity coefficient of material; denotes the initial velocity of impact; denotes the strain rate sensitivity coefficient of material; denotes the height compensation factor. Based on the difference between the effective compression height and the theoretical compression height is calculated based on simulation and experiment,choose between ,we choose [47].
The maximum effective EA of the Y-cell was:
here, denotes the height of the Y-cell.
Substituting Eqs. (1), (3) and (4) into Eq. (5), the maximum effective EA of the Y-cell can be obtained as follows:
[46] Li, M. Research on energy absorbers of Legged-type Lander and Dynamic simulation on its soft-landing process. Harbin Institute of Technology. 2013.
4. Point 13: the nature of the FEM model is still unknown - the pictures of deformed models suggest detailed shell model, however the foam material model should be rather applied to the solid model. Could Authors further explain this issue?
It has been corrected in the article.
In order to obtain more accurate compression rate parameters of three aluminum honeycomb materials (HB1, HB2 and HB3) under static pressure. According to the geometric parameters of HB1, HB2 and HB3 in Table 1 and the test sample structure in Figure 6, their shell model was established by using commercial software Workbench. The material density was set to , and ,respectively. The tensile strength was set to , and , respectively. The yield strength was set to and , respectively. The tetrahedral mesh with the mesh size of was selected. The bottom surface of the honeycomb material was fixed, the upper surface was applied with a pressure of , and the simulation time is set to . Simulation results of static pressure are shown in Figure 4 and Table1.
Reviewer 3 Report
Accept in the present format.
Thank you very much for your comments. These comments not only improve my writing, but also put forward many valuable opinions and Suggestions on my research field.